# α-catenin switches between a slip and an asymmetric catch bond with F-actin to cooperatively regulate cell junction fluidity

C. Arbore[1,2,6], M. Sergides [1,2,3,6], L. Gardini [1,4], G. Bianchi[1,2], A. V. Kashchuk[1,2], I. Pertici [5], P. Bianco [5], F. S. Pavone[1,2,4] & M. Capitanio [1,2✉]

α-catenin is a crucial protein at cell junctions that provides connection between the actin cytoskeleton and the cell membrane. At adherens junctions (AJs), α-catenin forms hetero-dimers with β-catenin that are believed to resist force on F-actin. Outside AJs, α-catenin forms homodimers that regulates F-actin organization and directly connect the cell membrane to the actin cytoskeleton, but their mechanosensitive properties are inherently unknown. By using ultra-fast laser tweezers we found that a single α-β-catenin heterodimer does not resist force but instead slips along F-actin in the direction of force. Conversely, the action of 5 to 10 α-β-catenin heterodimers together with force applied toward F-actin pointed end engaged a molecular switch in α-catenin, which unfolded and strongly bound F-actin as a cooperative catch bond. Similarly, an α-catenin homodimer formed an asymmetric catch bond with F-actin triggered by protein unfolding under force. Our data suggest that α-catenin clustering together with intracellular tension engage a fluid-to-solid phase transition at the membrane-cytoskeleton interface.

[1] LENS - European Laboratory for Non-linear Spectroscopy, University of Florence, Via Nello Carrara 1, 50019 Sesto Fiorentino, Italy. [2] Department of Physics and Astronomy, University of Florence, Via Sansone 1, 50019 Sesto Fiorentino, Italy. [3] Department of Physics, University of Cyprus, P.O. Box 20537 Nicosia 1678, Cyprus. [4] National Institute of Optics - National Research Council, Largo Fermi 6, 50125 Florence, Italy. [5] Department of Biology, University of Florence, Via Madonna del Piano 6, 50019 Sesto Fiorentino, Italy. [6] These authors contributed equally: C. Arbore, M. Sergides. ✉email: capitanio@lens.unifi.it

In any living cell, an array of mechanotransduction proteins responds to mechanical cues to trigger complex molecular signaling, driving cell morphology and gene expression profiles[1]. Among these proteins, α-catenin has been reported to act as a mechanosensor that regulates adherens junctions (AJ) in response to mechanical cues from neighboring cells in a tissue[2]. It is widely accepted that α-catenin in AJ binds β-catenin, which, in turn, connects to the cytoplasmatic portion of E-cadherin to constitute the cadherin-catenin complex (CCC). Since α-catenin also binds F-actin, α-catenin has been indicated as a major candidate for providing a link between the CCC and the actin cytoskeleton. This molecular link is required to provide mechanical stability to AJ. According to the most accepted model of α-catenin mechanosensitivity, tension on the actin cytoskeleton is transmitted to bound α-catenin inducing a conformational change that opens up (i.e. unfolds) its central M-domain and increases the bond strength[2]. α-catenin unfolding is predicted to expose cryptic binding sites for vinculin and other actin-binding proteins, which further reinforce the connection to the actin cytoskeleton. However, the role of α-catenin as the mechanical link between the CCC and F-actin has been challenged by the finding that the binding of mammalian α-catenin to F-actin and β-catenin is mutually exclusive[3].

Using a *Danio Rerio* α-catenin[4], Buckley et al. recently demonstrated that the bond between CCC and F-actin is reinforced by increasing force (i.e., it is a catch bond)[5]. However, it remains to be determined whether mammalian α-β-catenin heterodimers show similar catch-bond properties. Moreover, it is still unclear whether a single α-β-catenin heterodimer can bind to F-actin and bear force on it. CCCs have been recently shown to be organized in clusters in cells[6]. This finding raises the intriguing question whether CCC clustering and the interaction of F-actin with multiple α-catenins might be important in defining the mechanosensitivity of AJ.

Recent reports revealed the importance of α-catenin outside AJ as well, where it regulates F-actin organization[7] and promotes cell adhesion and migration by providing a direct link between the cell membrane and the actin cytoskeleton[8]. While α-catenin forms α-β-catenin heterodimers in AJ, outside AJ α-catenin binds itself and is prevalently found as homodimers. Despite its biological relevance, no data is currently available on the force-dependence of the interaction between α-catenin homodimers and F-actin.

Finally, it is still unclear whether α-catenin unfolds during its interaction with F-actin. Using a truncated α-catenin that lacked the actin and β-catenin binding domains, Yao et al. showed that forces of about 5 pN trigger reversible unfolding of the α-catenin M-domain and allow vinculin binding[9]. However, there is no direct evidence that forces applied to α-β-catenin heterodimers or α-catenin homodimers through F-actin are sufficient to induce α-catenin unfolding.

Here, we use ultrafast force-clamp spectroscopy, a technique with sub-millisecond and sub-nanometer resolution based on laser tweezers[10–13], to dissect the mechanosensitivity of single mammalian α-catenin homodimers and α-β-catenin heterodimers. We show that a single mammalian α-β-catenin heterodimer does not bear force on F-actin, but instead slips along F-actin in the direction of force. Conversely, the cooperative action of multiple molecules together with force applied toward F-actin pointed end triggers α-catenin unfolding and engages a catch bond with actin. Similarly, a single α-catenin homodimer forms a directionally asymmetric catch bond with F-actin triggered by α-catenin unfolding, which resists forces directed toward F-actin pointed end.

## Results

### A single α-catenin homodimer unfolds to form a catch bond with F-actin.
We first analyzed single purified recombinant α-catenin homodimers (named as α-catenin below). A single actin filament was suspended between two optically trapped beads and brought in close proximity to a single α-catenin that was attached to a micron-sized glass bead on the coverslip surface (Fig. 1a, Methods section). A constant force was applied to the bead-actin-bead complex (herein named "dumbbell") through a double force-clamp system. The applied force caused the dumbbell to move through the assay solution at a constant velocity and the force was alternated in direction to maintain the dumbbell within few hundred nanometer excursion (Fig. 1b, "1. Unbound"). When α-catenin bound to the actin filament and became loaded with the applied force, the dumbbell motion stopped within ~30 μs (Fig. 1b, "2. Bound"). The interactions between α-catenin and the actin filament were detected through the change in the dumbbell velocity and their lifetimes and position assessed under forces in the range of ± 17 pN (see Methods).

The lifetime of the interactions visibly increased with force (Fig. 1c and Fig. 1d–f). Plot of the event lifetime versus force showed that the bond between α-catenin and F-actin is reinforced by increasing force and responds asymmetrically with respect to the force direction (Fig. 1c). Data reported in Fig. 1 were acquired on the same dumbbell in order to maintain the actin filament orientation and reveal asymmetries in the force-dependence. Event lifetime at 2.3 pN (7.5 ± 0.8 ms) increases about 19-fold at 8.1 pN (145 ± 16 ms), reduces to 9.5 ± 2.6 ms at 11.5 pN and shows a second maximum at 13.8 pN (61 ± 11 ms). A similar effect was observed for force applied in the opposite direction, although the bond lifetime was about 7-fold shorter in the 0–8 pN range and with maximum lifetime at slightly different force values. Event lifetime at −4.3 pN (1.25 ± 0.07 ms) increases about 16-fold at −7.7 pN (20.5 ± 2.9 ms), reduces to 9.0 ± 1.8 ms at −8.6 pN, and shows a second maximum at −10.3 pN (42 ± 6 ms). Overall, these results show that a single α-catenin forms an asymmetric catch bond with F-actin that resists forces below 15 pN. The presence of two peaks in the interaction lifetime suggests the emergence of two conformational changes of α-catenin induced by force. Accordingly, we could fit each peak in the lifetime distribution with a two-state catch-bond model in which force triggers switching between a weak-binding and a strong-binding state (green and blue curves in Fig. 1c; see also methods and Supplementary Fig. 1). The distance parameters for the transitions between the two states indicate that large conformational changes (> 10 nm) occur during these transitions (Supplementary Table 1).

Analysis of α-catenin binding position at different forces further supports this view. Observation of position records showed that interactions occurred at single locations for forces below ~3-4 pN (Fig. 1d and Supplementary Fig. 2a), whereas above ~5 pN most interactions showed "steps" (red arrowheads in Fig. 1e,f and Supplementary Fig. 2b,c). This behavior raises the question whether the observed steps were a consequence of (i) α-catenin detaching and rapidly reattaching to the actin filament or (ii) α-catenin undergoing large conformational changes (i.e., unfolding) under force[9], as suggested by the analysis of the interaction lifetime. The former hypothesis would imply that steps occurred only in the direction of the force, following the periodicity of the actin filament. However, analysis of step size distribution at about +5 pN showed a main peak in the force direction centered around 12.1 ± 5.8 nm and the presence of a step in the opposite direction of similar amplitude (−11.2 ± 5.1 nm) (Fig. 1h). Likewise, step size distribution at −5 pN showed steps of similar amplitude in both directions (−11.4 ± 6.4 nm and 10.3 ± 4.2 nm) (Fig. 1g). Analysis of individual records around 5 pN force display α-catenin jumping back and forth between two position levels separated by about 10–20 nm (Supplementary Fig. 2d). This indicates that, around

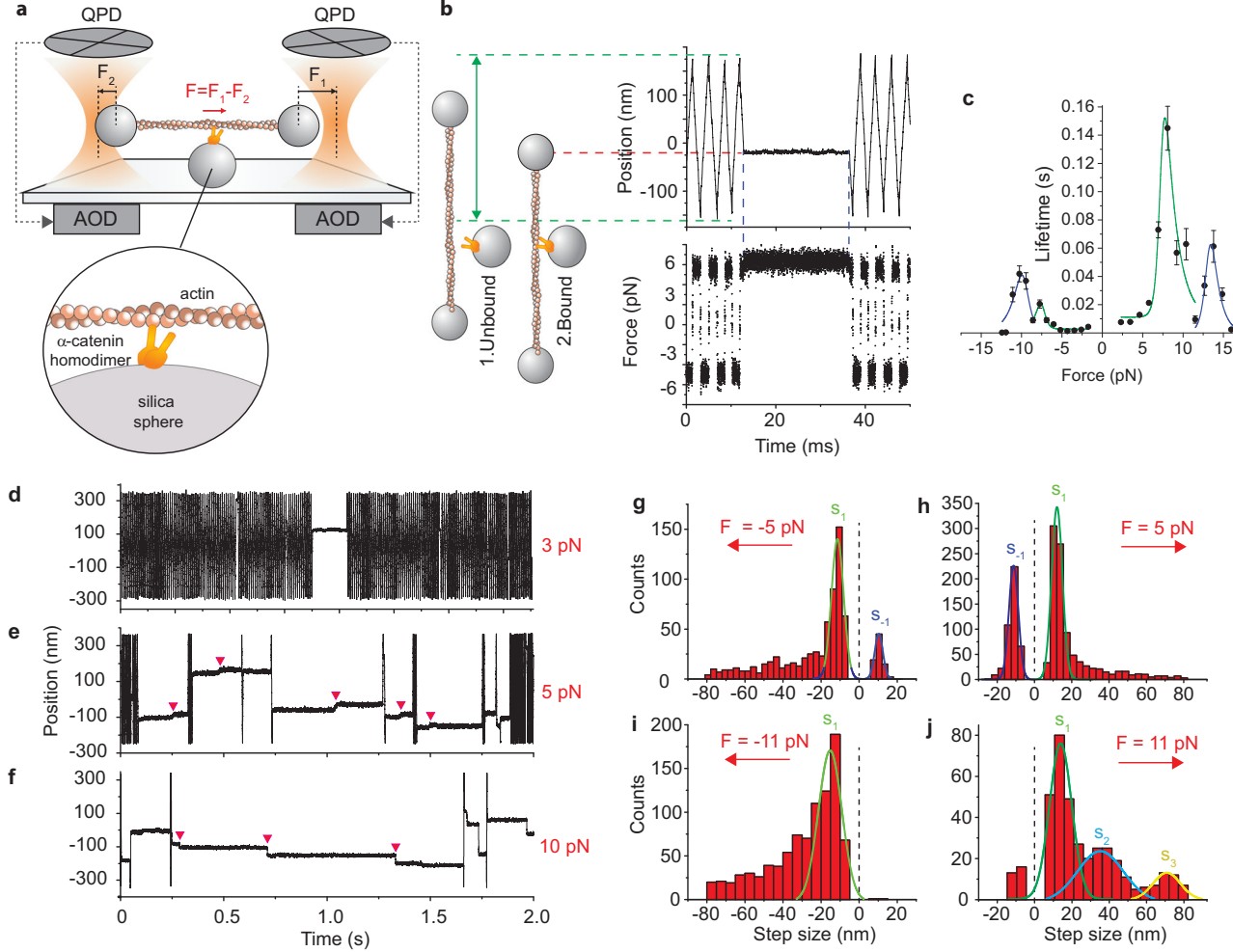

**Fig. 1 A single α-catenin homodimer unfolds and resists force on F-actin. a** Schematic of ultrafast force-clamp spectroscopy applied to a single α-catenin homodimer. An α-catenin molecule is attached to a glass bead stuck on the coverslip. A single α-catenin homodimer is composed by two His(6x) tagged α-catenins (see methods). An actin filament is suspended between two beads trapped in optical tweezers. Black arrows are the force clamped on the right ($F_1$) and left bead ($F_2$), red arrow is the net force ($F = F_1 - F_2$) on the dumbbell. $F$ is alternated in direction to maintain the dumbbell within a limited excursion. **b** Left: illustration showing the position of the right bead when: (1) α-catenin is unbound and the dumbbell oscillates; (2) α-catenin binds to the actin filament. Right: example trace showing displacement and force ($F \sim 6$ pN) during the corresponding phases of dumbbell oscillation and α-catenin attachment, under positive and negative loads. **c** Load-dependent lifetime of the interaction between a single α-catenin homodimer and F-actin. Green and blue lines are, respectively, fit of the low-force and high-force peak with the two-state catch-bond model (see methods). $n = 23234$ total number of interactions for all points in the plot. Error bars, s.e.m. **d–f** Example traces under force of about 3 pN, 5 pN, and 10 pN, respectively. Red arrowheads indicate steps occurring during the interaction. **g–j**, Distribution of step size at various forces. Gaussian fits of the main peaks (green, blue, and yellow lines) are also shown with their center and s.d. **g** $F = -5$ pN, $s_1 = -11.3 \pm 6.4$ nm, $s_{-1} = 10.3 \pm 4.3$ nm **h** $F = +5$ pN, $s_1 = 12.1 \pm 5.8$ nm, $s_{-1} = -11.2 \pm 5.1$ nm **i** $F = -11$ pN, $s_1 = -15 \pm 12$ nm **j** $F = +11$ pN, $s_1 = 14 \pm 12$ nm, $s_2 = 35 \pm 24$ nm, $s_3 = 71 \pm 15$ nm. Error, s.d. Source data are provided as a Source Data file.

5 pN, α-catenin is near the equilibrium between the folded and the unfolded state and switches between the two states because of thermal noise. Our results are in very good agreement with those from Yao et al., who measured a reversible unfolding step on the α-catenin M-domain occurring around the same force and with similar amplitude[9]. At higher force (±11 pN), the negative step is largely reduced (Fig. 1j) or disappears (Fig. 1i and Supplementary Fig. 3), confirming the idea that force drives the unfolding of the protein in the force direction and that the reverse folding step is inhibited at large forces. Moreover, multiple steps and larger irreversible steps of about 35 and 71 nm become apparent in the position records (Fig. 1f and Supplementary Fig. 2c) and in the step distributions (Fig. 1j) in the direction of the force. These larger steps are compatible either with additional irreversible unfolding steps as observed by Yao et al. at about 12 pN[9], or from rapid unbinding-rebinding of α-catenin from the actin filament, although the presence of the above mentioned two peaks in the

interaction lifetime supports the former hypothesis. Moreover, possible contributions to the step size distribution due to the simultaneous binding of the two α-catenin monomers to actin are discussed in the supplementary discussion.

**A single α-β-catenin heterodimer forms a slip bond with F-actin**. We then analyzed the interaction between a single α-β-catenin heterodimer and an actin filament (Fig. 2a). Plot of the event lifetime versus force showed that the bond between a single α-β-catenin heterodimer and F-actin does not bear force (Fig. 2b). The bond lifetime decreased with force, occurring in the millisecond time scale at forces around 5 pN and rapidly dropping below 1 ms for $F > 7$ pN. We could detect such rapid interactions thanks to the unprecedented time resolution of our technique[10]. The lifetime vs force plot was well fitted by the Bell slip-bond model, in which the lifetime decreases exponentially with force

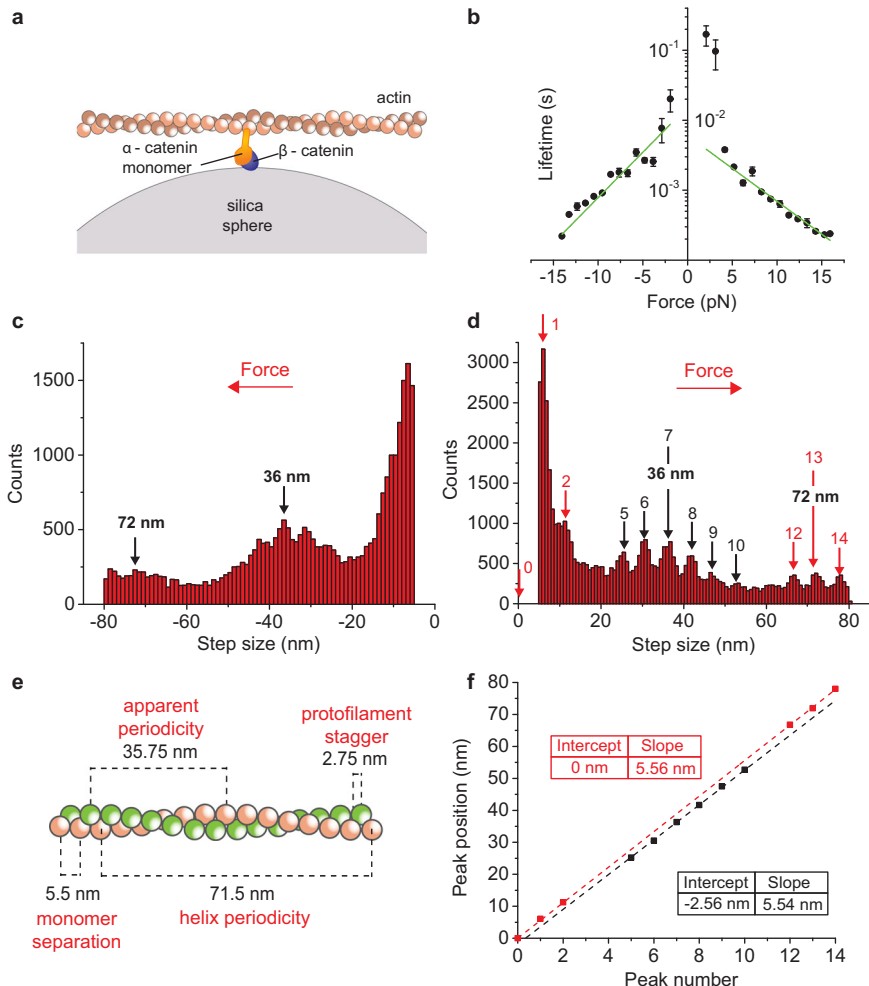

**Fig. 2 A single α-β-catenin heterodimer forms a slip bond with F-actin that does not bear force. a** Schematic of ultrafast force-clamp spectroscopy applied to a single α-β-catenin heterodimer. The single heterodimer is composed by a single monomeric His(6x) tagged α-catenin attached onto a layer of GST-tagged β-catenin (see methods). **b** Load-dependent lifetime of the interaction between a single α-β-catenin heterodimer and F-actin. Green lines are fits of lifetimes $\tau$ with the Bell-bond equation $\tau = \tau_0^{\pm} \exp\left(-\frac{d_{\alpha\beta}^{\pm}F}{k_BT}\right)$ (see Methods). Fitting parameters are $d_{\alpha\beta}^{+} = 0.88 \pm 0.05$ nm, $d_{\alpha\beta}^{-} = 1.22 \pm 0.08$ nm, $\tau_0^{+} = 6.1 \pm 0.8$ ms, $\tau_0^{-} = 16 \pm 4$ ms. $n = 360486$ total number of interactions for all points in the plot. Error bars, s.e.m. **c**, **d** Distribution of step size vs. force for negative (**c**) and positive force (**d**). **e** Illustration showing the actin filament structure and highlighting the two protofilaments (red and green), the helix and apparent periodicity, the distance between consecutive monomers on the same protofilament, and the stagger between monomers on two consecutive protofilaments. **f** Plot of the position of peaks obtained from the step histogram in (**d**) against peak number. Red squares correspond to monomers on the same protofilament. Black squares correspond to monomers on the adjacent protofilament. Red and black dashed lines are the least-squares fittings of the red and black points, giving slopes of 5.56 ± 0.01 and 5.54 ± 0.04 nm/monomer respectively, and stagger between protofilaments of 2.56 ± 0.31 nm. Errors, s.e.m. Source data are provided as a Source Data file.

(green lines in the log plot of Fig. 2b). Position records show that interactions of the α-β-catenin heterodimer also included "steps" (Supplementary Fig. 4). However, the step size distribution was very different than in the case of the α-catenin homodimer, displaying (i) negligible number of negative steps and (ii) strong 5.5 nm periodicity, which is characteristic of the actin monomer separation[14,15] (Fig. 2c,d and Supplementary Fig. 5). The actin filament is made of two protofilaments composed by 13 monomers per helix turn, giving a periodicity of 13 × 5.5 nm = 71.5 nm. The two protofilaments are separated by a stagger of 2.75 nm, giving an apparent periodicity of 6 × 5.5 nm + 2.75 nm = 35.75 nm (Fig. 2e). Figure 2f shows the peak position, obtained from the step size distribution in Fig. 2d, against the peak number, as defined in Fig. 2d. Linear regression analysis showed that the points corresponding to peaks 0–2 and 12–14 (red squares) and those corresponding to peaks 5–10 (black squares) lied on two parallel lines with nearly identical slopes (5.56 ± 0.01 and 5.54 ± 0.04 nm/peak for the red and black

lines respectively), but displaced by 2.56 ± 0.31 nm. The line slopes clearly corresponded to the distance between contiguous actin monomers, while the distance between the two lines corresponded to the protofilament stagger.

These results strongly indicate that the observed steps are due to a single α-catenin that rapidly unbinds from an actin monomer, slips along the actin filament in the direction of the force, and rebinds to a neighbor actin monomer. Moreover, no apparent unfolding was detected. In fact, all peaks corresponded to the actin monomer separation while a negligible number of negative steps was detected. Inspection of records at any force did not show α-catenin jumping back and forth between two position levels as in the case of the α-catenin homodimer. Overall, these results indicate that a single α-β-catenin heterodimer forms a slip bond with F-actin that does not resist force. Force applied by the actin filament to the α-β-catenin complex causes a rapid slip to a neighbor actin monomer rather than inducing α-catenin

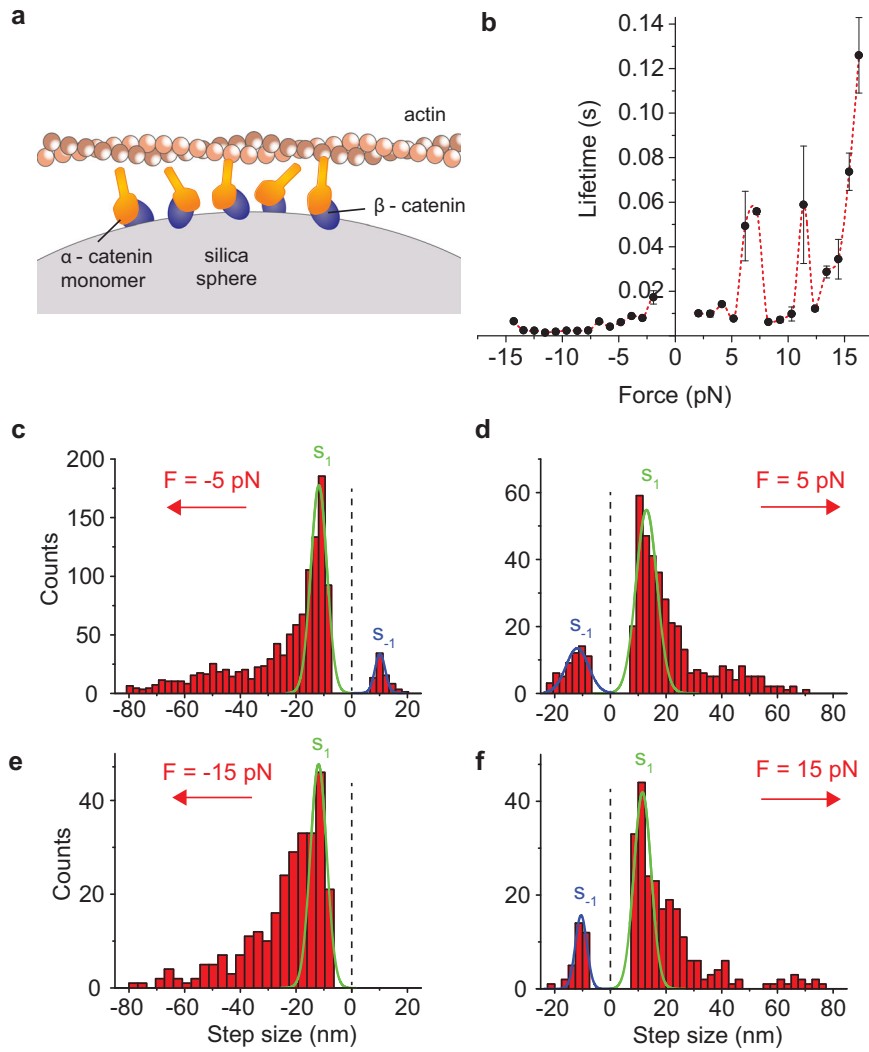

**Fig. 3 Multiple α-β-catenin heterodimers restore force-bearing capability on F-actin. a** Schematic of ultrafast force-clamp spectroscopy applied to multiple α-β-catenin heterodimers. Multiple heterodimers are composed by monomeric His(6x) tagged α-catenins attached onto a layer of GST-tagged β-catenin (see methods). **b** Load-dependent lifetime of the interaction between multiple α-β-catenin heterodimers and F-actin. $n = 55171$ total number of interactions for all points in the plot. Error bars, s.e.m. **c–f** Distribution of step size at various forces. Gaussian fits (green and blue lines) of the main peaks are also shown with their center and s.d. **c** $F = -5$ pN. $s_1 = -11.7 \pm 2.9$ nm, $s_{-1} = 10.2 \pm 1.8$ nm. **d** $F = +5$ pN, $s_1 = 12.8 \pm 3.8$ nm, $s_{-1} = -12.1 \pm 4.0$ nm. **e** $F = -15$ pN, $s_1 = -11.9 \pm 2.9$ nm. **f** $F = +15$ pN, $s_1 = 11.6 \pm 3.1$ nm, $s_{-1} = -10.5 \pm 2.1$ nm. Error, s.d. Source data are provided as a Source Data file.

unfolding and strong binding to actin. The slip-bond does not show an asymmetric response to force, as data reported in Fig. 2 were acquired as before on the same dumbbell to maintain the actin filament orientation. Possible experimental artifacts due to non-specific interaction of the actin filament with the surface and/or to the presence of a His(6x) tag in the α-catenin molecule and a GST tag in the β-catenin molecule have been carefully considered and are discussed in the supplementary methods and discussion.

**Multiple α-β-catenin heterodimers form a cooperative catch bond with F-actin.** Since single mammalian α-β-catenin complexes cannot bear force on actin, we questioned whether the cooperative action of multiple complexes might resist the physiological forces that occur at AJ. We, thus, increased the α-β-catenin heterodimer concentration 10-fold to have multiple proteins interacting simultaneously with the actin filament (Fig. 3a). Similarly to what we observed using α-catenin homodimers, the bond between α-catenin and F-actin was reinforced

by increasing force and responded asymmetrically with respect to the force direction, with lifetime peaks centered about 7.2 pN and 11.4 pN (Fig. 3b). Moreover, an additional peak was detected at the highest force that we could probe (~17 pN). Also here, data were acquired on a single dumbbell to keep the actin filament orientation fixed and reveal asymmetries in the force-dependence. The asymmetric response to force was more pronounced here, resulting in a catch bond for force applied in one direction and a slip bond in the opposite direction (Fig. 3b and Supplementary Fig. 6). Position records were also similar to the α-catenin homodimers, mainly composed of single events at forces below ~3 pN and multiple "steps" above ~5 pN (Supplementary Fig. 7). Step size distribution at about −5 pN showed a main peak centered around −11.7 ± 2.9 nm in the force direction and the presence of a step in the opposite direction of similar amplitude (10.2 ± 1.8 nm) (Fig. 3c). Similarly, step size distribution at about +5 pN showed a main peak centered around 12.8 ± 3.8 nm and the presence of a negative step of similar amplitude (−12.1 ± 4.0 nm) (Fig. 3d). Analysis of individual records at force greater than 5 pN often displayed α-catenin jumping back and

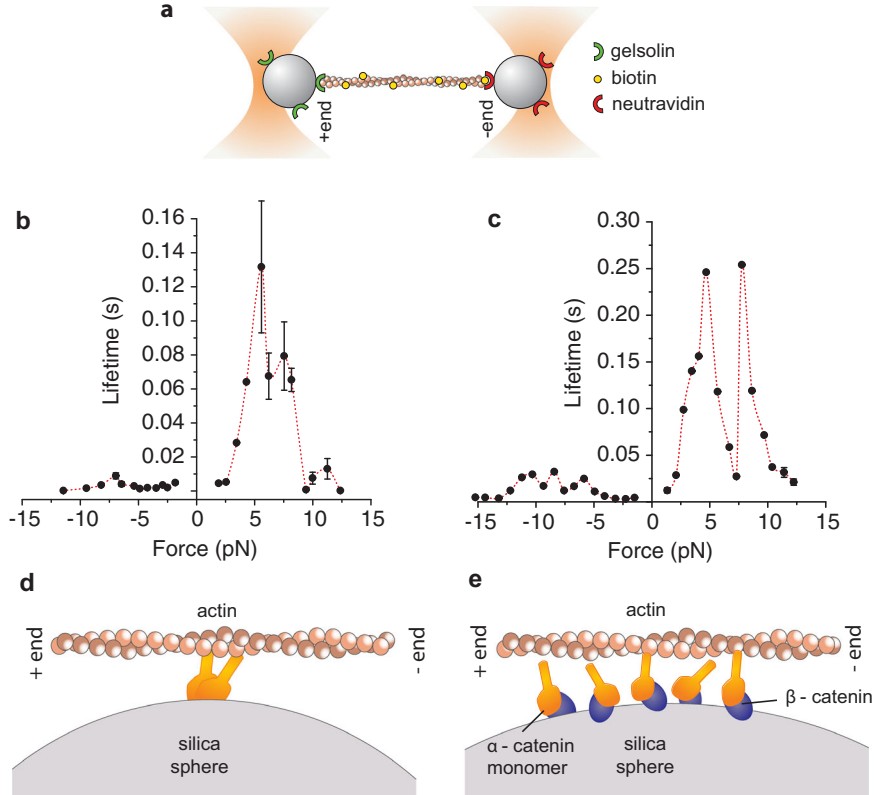

**Fig. 4 α-catenin resists load directed toward the F-actin pointed end. a** Schematic of the oriented actin filament experiments. **b** Load-dependent lifetime of the interaction between a single α-catenin homodimer and F-actin. Average from 3 dumbbells. $n = 9891$ total number of interactions for all points in the plot. Error bars, s.e.m. **c** Load-dependent lifetime of the interaction between multiple α-β-catenin heterodimers and F-actin. Average from 3 dumbbells. $n = 79790$ total number of interactions for all points in the plot. Error bars, s.e.m. Error bars are smaller than symbols. Source data are provided as a Source Data file. **d**, **e** Schematic of the single α-catenin homodimer (**d**) and multiple α-β-catenin heterodimers (**e**) interacting with an oriented actin filament.

forth between two position levels separated by 10–20 nm (Supplementary Fig. 7). The negative step rapidly disappeared at increasing negative forces in which the bond slips, but it was still present at the highest positive forces in which the bond lifetime is maximum (Fig. 3f and Supplementary Fig. 8b).

**The bond between α-catenin and F-actin is asymmetric and stronger for loads directed toward the F-actin pointed end.** Since the lifetime vs force response for both a single α-catenin homodimer and multiple α-β-catenin heterodimers was strongly asymmetric, we developed an experimental assay to determine the polarity of the actin filament in each dumbbell. To this end, in one of the two traps we grabbed an actin-tailed bead in which the barbed end (+) of a biotinylated actin filament was specifically linked to a gelsolin-coated bead. In the other trap, we caught a neutravidin-coated bead. Actin on the gelsolin-coated beads was rhodamine-labelled, whereas neutravidin-coated beads were labelled with Alexa647 to clearly distinguish the two beads and assess the actin filament orientation (Fig. 4a). Using the oriented dumbbell, we repeated measurements on a single α-catenin homodimer and multiple α-β-catenin heterodimers and analyzed the load-dependence of the interaction lifetimes. Since the orientation of actin in each experiment was determined, we could average data acquired from different dumbbells and molecules. Figures 4b and 4c show the load dependence of the interaction lifetime for a single α-catenin homodimer and multiple α-β-catenin heterodimers, respectively. Data clearly show that, in both cases, the interaction lifetime is greatly enhanced when force is applied toward the actin minus end.

## Discussion

Here, we provided direct evidence that a single mammalian α-catenin bound to β-catenin forms a slip bond with F-actin. The weak and rapid interactions that we observed could be detected thanks to the 10–100 μs time resolution of our setup, which is about two orders of magnitude better than conventional optical trapping assays. The limited-time resolution is likely the reason why previous reports could not detect the slip bond between a single α-β-catenin heterodimer and F-actin[5], whereas they could possibly detect the action of multiple molecules, which behave as a catch bond and produce much longer interactions than single molecules under force.

Our results indicate that α-catenin binds F-actin as an unconventional slip bond that cooperatively switches into an asymmetric catch bond. A cooperative action of multiple α-catenin molecules is required because a mere sum of multiple slip bonds would result again in a slip bond. Further insight into this mechanism can be gained by analysis of the duty ratio $r$, i.e., the fraction of time that a molecule spends attached to the actin filament: $r = \tau_{on}/(\tau_{on} + \tau_{off})$[16]. When the number of α-catenin molecules that can interact with the actin filament is above $N_{min} = 1/r$, at least one molecule is bound to actin on average and bear force on it (see Methods). The duty ratio of a single α-catenin homodimer rapidly switches from $r < 0.2$ for low forces ($F < 5$ pN) to $r = 0.8$–$0.9$ in the force range where also the lifetime is maximized ($5 < F < 15$ pN, Fig. 5a). Therefore, α-catenin homodimers under force remain bound to actin most of the time and an average number of 1.1–1.2 molecules can sustain force on an actin filament for a prolonged time. On the contrary, the duty ratio of a single α-β-catenin heterodimer is roughly

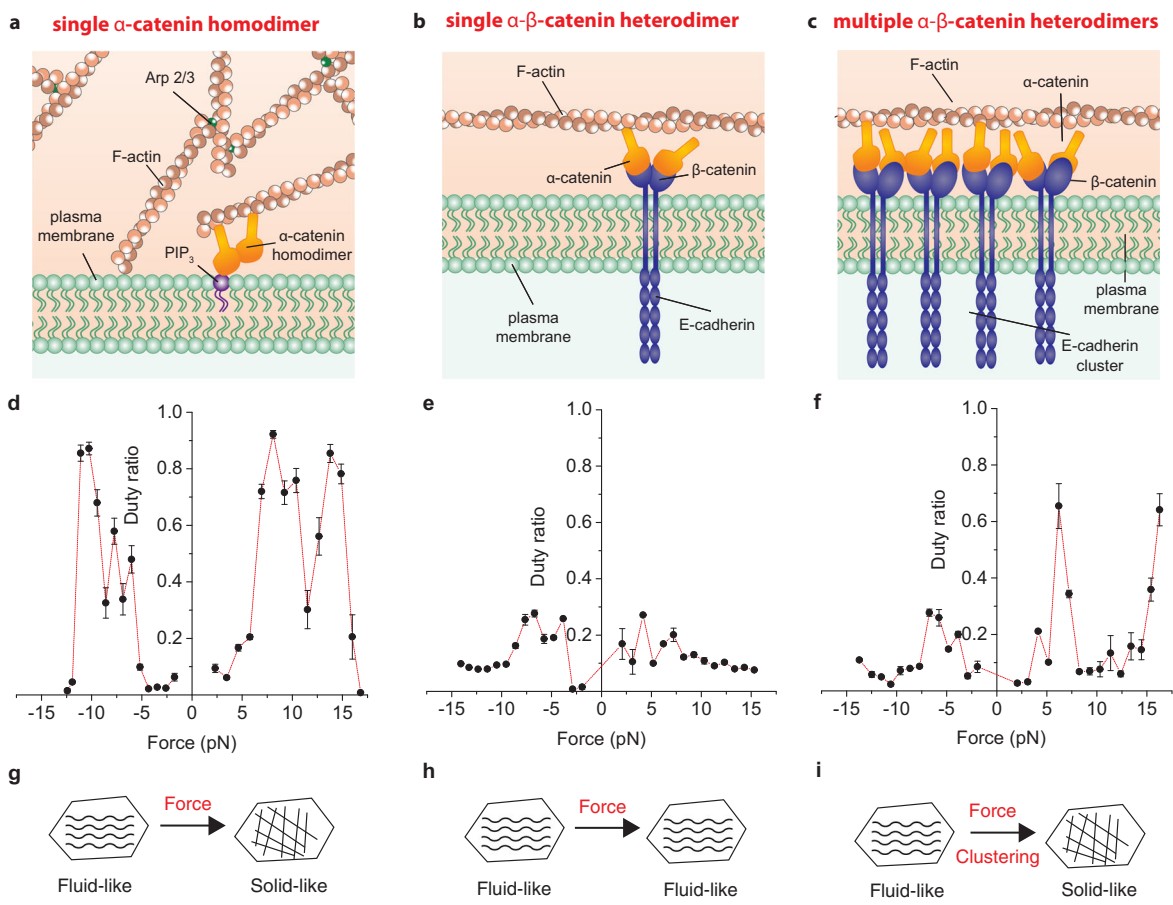

**Fig. 5 α-catenin clustering and intracellular tension regulate cell junction fluidity. a** Illustration showing α-catenin in filopodia. α-catenin is present in the form of homodimers bound to PIP$_3$ on the plasma membrane. **b** Illustration showing α-catenin in immature adherens junctions. α-catenin is present in the cadherin-catenin complexes at low density. **c** Illustration showing α-catenin in mature adherens junctions. α-catenin is present in the cadherin-catenin complexes which cluster in groups of 5–6. **d** Duty ratio of a single α-catenin homodimer. Forces above 5 pN induce α-catenin unfolding and large increase in duty ratio. $n = 23234$ total number of interactions for all points in the plot. Error bars, s.e.m. **e** Duty ratio of a single α-β-catenin heterodimer is not affected by force. $n = 360486$ total number of interactions for all points in the plot. Error bars, s.e.m. **f** Duty ratio of multiple α-β-catenin heterodimers. Forces above 5 pN induce α-catenin unfolding and large increase in duty ratio. $n = 55171$ total number of interactions for all points in the plot. Error bars, s.e.m. **g** Interaction between a single α-catenin homodimer and F-actin switches from a fluid to a solid response under force. **h** Interaction between a single α-β-catenin heterodimer and F-actin is fluid regardless of the tension on the actin cytoskeleton. **i** Interaction between multiple α-β-catenin heterodimers and F-actin switches from a fluid to a solid response under force. Source data are provided as a Source Data file.

constant to $r \sim 0.1$–$0.2$ independently of force (Fig. 5b), whereas the cooperative action of multiple molecules is required to increase the duty ratio to $r \sim 0.7$ under directional force (Fig. 5c). The $r \sim 0.1$–$0.2$ for a single α-β-catenin heterodimer indicates that there should be about 5–10 molecules available for binding to actin to maintain continuous contact with the actin filament. Strikingly, CCC in AJ have been found to cluster in nanodomains composed by 5–6 molecules[17]. Our data can, thus, provide an explanation of the biological relevance of CCC clustering[6,17–19]. In fact, below 5–10 molecules the probability of finding multiple molecules bound simultaneously to an actin filament is small. The engagement of the cooperative catch bond mechanism observed here is unlikely, and AJ would not be able to bear force on actin. Previous studies indicate that α-catenin binds cooperatively to F-actin by altering the actin filament conformation[20], which might be the molecular mechanism at the base of the change in the load-dependence from single to multiple α-β-catenin heterodimers that we observed. Moreover, such asymmetric cooperative load-dependence might itself (i) cause unidirectional flow of CCC in cell-cell junctions[21] and (ii) drive cluster formation through increased diffusional trapping mediated by α-

catenin cooperativity and tension applied by myosin motors on the actin cytoskeleton[6].

An indication of the average number of α-catenins that were simultaneously bound to actin in the different experiments can be obtained from a measurement of the average stiffness of the molecules bound to the dumbbell during the interactions (see methods). The single α-β-catenin heterodimer gave an average stiffness of $0.263 \pm 0.006$ pN/nm (error s.e.m), about half of the α-catenin homodimer ($0.46 \pm 0.03$ pN/nm). Assuming that the stiffness of an α-catenin monomer in the homodimer and in the heterodimer is the same, this result indicates that both α-catenins in the homodimer participate to the interaction with actin for most of the time ($\sim 87\%$ from the ratio of the homodimer stiffness to double the heterodimer stiffness). In the experiments with multiple α-β-catenin heterodimers, the average stiffness was in between the single heterodimer and the single homodimer ($0.336 \pm 0.007$ pN/nm), suggesting that simultaneous binding of two α-catenin molecules occurred for about 64% of the time in these experiments. This result reinforces our conclusion that the catch bond behavior is a consequence of the cooperative binding of multiple α-catenin molecules to actin.

Notably, our data show that the catch-bond behavior is accompanied by a large conformational change (unfolding) of the α-catenin molecule, both in a single α-catenin homodimer and in multiple α-β-catenin heterodimers. Unfolding is not observed in a single α-β-catenin heterodimer slip bond, indicating that α-catenin unfolding provides the molecular switch that converts the link to F-actin into a catch bond. The measured unfolding step size is compatible with unfolding of the bundled α-helices in the vinculin binding domain into a linear chain of extended α-helices[9]. The unique ultra-fast force-clamp configuration used here allowed us to directly observe under which conditions folding/unfolding transitions of the α-catenin molecule actually occur when force is applied to the interacting actin filament, as previously hypothesized[9].

Finally, the bond between actin and both α-catenin homodimers and multiple α-β-catenin heterodimers showed a strong asymmetry. The lifetime was much longer when force was applied toward the actin pointed end (Fig. 4). This result has strong biological relevance because, in cells, intracellular tension is produced toward the actin pointed end by non-muscle myosin and/or by the actin retrograde flow. The directionality of the link between α-catenin and actin can, thus, represent an important mechanism for the selective transmission of tension to adherens junctions. A similar mechanism can also be present in focal adhesions, as recent results show that α-catenin directly interacts with integrin adhesions to regulate their growth and transmission of mechanical forces into the matrix[22].

Based on our results, we developed a simple chemo-mechanical model to relate the sliding velocity $v$ of an actin filament interacting with α-catenin to the force applied to it ($F$), as a function of the α-catenin duty ratio ($r$) (see methods). When a constant force $F$ is applied to an actin filament immersed in a viscous fluid, the filament slides at constant velocity $F/\gamma$ against the viscous drag of the fluid $\gamma$, during the fraction of time when α-catenin molecules are detached. Therefore, the sliding velocity is obtained as:

$$v = \frac{\tau_{off}}{\tau_{on} + \tau_{off}} \cdot \frac{F}{\gamma} = (1 - r)\frac{F}{\gamma} \Longrightarrow F = \frac{\gamma}{1 - r}v \qquad (1)$$

This equation tells us that when the duty ratio is significantly smaller than one and independent of force, as in the case of a single α-β-catenin heterodimer (Fig. 5b), the α-catenin-actin interaction creates additional viscous drag on the actin filament, but the system behaves as a fluid ($F \propto v$). However, if the duty ratio gets closer to one, as it occurs when force directed toward the actin pointed end is applied to a single α-catenin homodimer (Fig. 5a) or to multiple α-β-catenin heterodimers (Fig. 5c), the velocity drops to zero and the system behaves as a solid. Therefore, α-catenin acts as a cooperative molecular switch that responds to tension to directionally regulate the fluidity of its link to actin. This result is particularly notable in light of recent findings of the occurrence of fluid-to-solid transitions in cultured epithelial monolayers[23] and in-vivo[24]. Notably, disruption of E-cadherin clusters in living cells has been shown to increase cell-cell junction fluidity[25]. We thus propose that, as a consequence of α-catenin cooperative mechanosensitivity, AJ might switch from a fluid phase in the absence of CCC clustering to a solid phase when CCC become organized into clusters and tension applied by non-muscle myosin or generated by the actin retrograde flow builds up on the actin cytoskeleton.

## Methods

**Protein expression and purification.** The pET28a(+) vector (Novagen) containing the full-length mouse αE-catenin was an offering from Dr. Noboru Ishiyama[26]. The plasmid encodes the protein fused with a 6x His-tag at the N-terminus with a linker sequence containing a thrombin cleavage site and restriction sites for NdeI and NheI (Amino acid sequence of the His tag and linker:

"MGSSHHHHHHSSGLVPRGHMAS"). The plasmid was transformed in BL21 DE3 pLysS strains (Agilent). Recombinant αE-catenin was expressed via isopropyl 1-thio-β-d-galactopyranoside (IPTG,Sigma-Aldrich) induction. Cultures were grown to an OD of 0.7 and induced with 0.5 mM IPTG for 3 h at 37 °C. Protein was purified via FPLC (AKTA prime system, GE Healthcare) using HisTrap HP columns and eluted in 25 mM Tris, pH 7.5, 300 mM NaCl, 300 mM Imidazole. Size exclusion chromatography was performed in 25 mM Tris, pH 7.5, 150 mM NaCl, 1 mM DTT through with a HiLoad Superdex 200 (GE, Healthcare) as a further purification step. Monomeric and homodimeric fractions were pooled separately and purity was evaluated by SDS-PAGE gel electrophoresis. Protein concentration was measured using absorbance at 280 nm (Supplementary Fig. 9). Human β-catenin, GST tagged was bought from Sigma-Aldrich (SRP5172). β-catenin is fused with a GST tag at the N-terminus with a linker sequence containing a thrombin digestion site (amino acid sequence of the linker: "SDLVPRGS"). G-actin and biotinylated actin were bought from Cytoskeleton, Inc. (AKL99, AB07-A). Actin filaments were obtained by incubating 0.4 μg·μl$^{-1}$ biotinylated actin and 0.2 μg·μl$^{-1}$ actin in 50 mM KCl, 2 mM MgCl$_2$ and 1 mM ATP in 10 mM Tris, pH 7.5 (actin polymerization buffer, Cytoskeleton, Inc. BSA02) for an hour. F-actin filaments were labeled overnight with 10 μM rhodamine-phalloidin and 20 mM DL-Dithiothreitol.

**Pull down and flow-cell assay.** For the flow-cell assay, a 1% nitrocellulose-pentyl acetate solution was smeared on a coverslip and assembled to make a flow chamber[27]. For reactions containing β-catenin, 80 μg/ml of protein was allowed to react with the surface for 5 minutes. αE-catenin was then incubated for 5 minutes to allow binding. Unbound protein was washed away with a solution containing 20 mM MOPS pH 7.2, 50 mM KCl, 1 mM MgCl$_2$, 0.1 mM EGTA and 1 mg/ml bovine serum albumin, following incubation with a solution containing 100 nM rhodamine phalloidin F-Actin in 20 mM MOPS pH 7.2, 50 mM KCl, 1 mM MgCl$_2$, 0.1 mM EGTA, 1.2 μM glucose-oxidase, 0.2 μM catalase, 17 mM glucose and 20 mM DTT. Flow-cell assays were performed on an inverted fluorescence microscope (Nikon ECLIPSE TE300) equipped with a 532 nm laser (Coherent Sapphire) for rhodamine excitation (~3 mW on the sample). Images were acquired in total internal reflection configuration, through Nikon Plan Apo TIRF, 1.45 oil immersion objective on an EMCCD camera (Andor, iXon X3) with 82 nm pixel size, and 40 × 40 μm$^2$ field of view. Integration time was 0.2 s, and EM gain 40.

For the pull-down assay of αE-catenin with F-actin, 3 μM F-actin was incubated with increasing concentrations of αE-catenin (0.05 μM–8 μM) in 20 mM MOPS pH 7.2, 50 mM KCl, 0.1 mM EGTA, 1 mM MgCl$_2$. Binding reactions were performed at room temperature for 20 min and spun at 14000 rpm for 2 h at 4 °C. The supernatant was collected and the pellet resuspended in the original volume of assay buffer. Laemmli Sample Buffer (Sigma-Aldrich) was added to the samples and pellet and supernatant fractions were run on 4–12% BIS-Tris gels (Thermo Fisher Scientific). The gels were stained with Brilliant Blue reagent (Sigma-Aldrich) and scanned with a ChemiDoc$^{TM}$ MP Imaging system (Bio-Rad) and analyzed by densitometry with ImageJ.

For the pull-down assay of αE-catenin with β-catenin, we followed the protocol by Lapetina et al.[28] with the following modifications. β-catenin was conjugated to 1 μm diameter carboxyl-polystyrene beads 10% v/v (sigma-Aldrich) using the method established by Pertici et al.[29]. At the end of the first day of preparation, for the β-catenin conjugation, 40 μl of 200 μg/ml β-catenin was added to 4.8 μl functionalized beads and incubated overnight at 4 °C. In the second day of preparation, during the blocking step, 10 mg/ml of BSA (sigma-Aldrich) was added to 4.8 μl of β-catenin conjugated polystyrene beads. For control, we prepared BSA coated beads following Lapetina et al.[28]. β-catenin and BSA conjugated beads were incubated with increasing concentrations of αE-catenin (2 nM–150 nM) in 400 μl of binding buffer (25 mM HEPES, 500 mM KCl and 1 mM DTT). Beads concentration was adjusted to have β-catenin at a fixed concentration of 21.7 nM. Binding reactions were performed at room temperature for 30 min and the sample spun at 19400 rcf for 10 min at 4 °C. The supernatant was collected and the pellet resuspended in 30 μl of binding buffer. 7.5 μl Laemmli Sample Buffer (Thermo Fisher Scientific) was added to 22.5 μl of the supernatant and pellet fractions, which were run on 4–12% BIS-Tris gels (Thermo Fisher Scientific). The gels were stained with Brilliant Blue reagent (Sigma-Aldrich) and scanned with a ChemiDoc$^{TM}$ MP Imaging system (Bio-Rad) and analyzed using ImageJ.

**Optical trapping experiments.** Neutravidin beads were prepared as follows: 0.5% carboxylated microspheres (0.9 μm diameter, Sigma Aldrich) were incubated with 7.4 mg/ml EDC (N-(3-Dimethylaminopropyl)-N′-ethylcarbodiimide hydrochloride, Sigma-Aldrich) and 0.2 mg/ml biotin-x-cadaverine (Molecular Probes, A1594) for 30 minutes at room temperature. Biotinylated beads were then incubated with 0.1 M glycine, 0.5 mg/ml neutravidin and 0.02 mg/ml streptavidin-alexa532 or, alternatively, with 0.02 mg/ml streptavidin-alexa647 (Thermo Fisher Scientific) in the experiments with oriented dumbbells[30]. In the latter experiments, the correct orientation of the biotinylated actin filament was achieved by using the Bead Tailed Actin (BTA) method established by Pertici et al.[31]. The method takes advantage of the property of Gelsolin TL40 to cap the barbed (+end) of the actin filament. Gelsolin TL40 (N-terminal cytoplasmic gelsolin, Hypermol, Germany) was covalently bound to carboxylated microspheres (0.9 μm diameter, Sigma

Aldrich) with 1-ethyl-3-(3-dimethylaminopropyl)-carbodiimide in order to bind the +end of a single actin filament. TL40-coated beads were stored in the stock solution (150 mM NaCl, 20 mM sodium phosphate buffer pH 7.4, 0.1 mM ATP, 10 mg/ml BSA, 5% (v/v) glycerol, and 3 mM NaN₃) at 0 °C for about 6 months.

Silica beads (1.2 μm, diameter, Bangslabs), dispersed in 1% nitrocellulose-pentyl acetate solution, were smeared on a coverslip and allowed to dry before the flow chamber was assembled[27,30]. For reactions containing β-catenin, 80 μg/ml of protein was allowed to react with the surface for 5 min. αE-catenin (4 μg/ml for single molecule experiments or 40 μg/ml for multiple molecule experiments) was incubated for 5 min to form the α/β catenin complex, or in the absence of β-catenin to bind non-specifically the nitrocellulose surface. The flow chamber was then incubated with 1 mg/ml bovine serum albumin (Sigma-Aldrich) for 5 min to minimize non-specific interactions. In the experiments with non-oriented dumbbells, a solution containing the neutravidin coated-beads and the biotinylated actin was loaded into the chamber. The final reaction was composed by 0.005% neutravidin functionalized beads, 1 nM rhodamine phalloidin F-Actin in 20 mM MOPS pH 7.2, 50 mM KCl, 1 mM MgCl₂, 0.1 mM EGTA, 1.2 μM glucose-oxidase, 0.2 μM catalase, 17 mM glucose and 20 mM DTT. Measurements were carried out after sealing the flow chamber with high vacuum grease. A single biotinylated actin filament was anchored through specific binding to the trapped neutravidin-coated microspheres and pre-tensioned to ~3 pN. At low concentration of αE-catenin, approximately 1 in 4 silica beads showed interaction with the actin filament, providing evidence that the large majority of the beads contained at most a single catenin molecule.

Control experiments to probe non-specific interactions were performed without αE-catenin. Few tens of beads were tested in different slides and negligible non-specific interactions were detected in these conditions (supplementary discussion, section "Non-specific interactions"). Moreover, the experimental conditions assure the minimization of possible artifactual contributions to our measurements which might derive by the presence of a mixture of homodimers and heterodimers in the experiments (see supplementary discussion, section "Dimeric vs monomeric α-catenin").

In the experiments with oriented dumbbells, BTA (in a solution of 25 mM imidazole, 25 mM KCl, 4 mM MgCl₂, 1 mM DTT, 3 mM NaN₃) and neutravidin-coated beads were loaded into the chamber, in the same final reaction mix, and discriminated under fluorescence microscopy from the different emission spectra of the rhodamine phalloidin and alexa647, respectively. First, a BTA with an anchored actin filament was detected under 532 nm laser illumination and trapped. Second, switching to the 635 nm laser, a neutravidin-coated bead was identified and captured in the second trap. The biotinylated actin filament linked to the TL40 gelsolin coated bead was anchored through specific binding to the trapped neutravidin-coated bead and pre-tensioned to ~3 pN. The dumbbell orientation with respect to the two traps was changed from one experiment to another to exclude that trap asymmetries might influence the results. The experimental conditions and control experiments for the oriented dumbbell were otherwise the same described above for the non-oriented dumbbell experiments.

**Ultrafast force-clamp spectroscopy setup.** Ultra-fast force-clamp spectroscopy and the experimental apparatus are described in detail elsewhere[10,30]. In brief, dual optical tweezers and fluorescence microscopy were combined in a custom inverted microscope. The two laser traps were independently controlled by two acousto-optic deflectors (AODs). The forward scatter signal from the tweezers was filtered by two interferential filters before reaching the two quadrant detector photodiodes (QDPs) for particle position measurements. The applied force was calculated from the displacement of the bead (x) and the trap stiffness (k). The trap stiffness was always calibrated prior experiments for the entire range of the trap positions using a power spectrum method[32]. A high-magnification camera (CCD 2000X) monitored the coordinates of the pedestal bead and with the aid of a 3D piezo translator system nm-stabilization against thermal drifting and low-frequency noise was achieved[33]. Additionally, a 532 nm laser (Coherent Sapphire) was used for the excitation of rhodamine-labelled F-actin and streptavidin-alexa532 coated beads. A 635 nm laser (Blue Sky Research, FTEC2635-P60SF30) was used for the excitation of streptavidin-alexa647 coated beads. Both data acquisition and force-clamp feedback were controlled at 200 kHz sample rate through an FPGA board (NI-PCI-7830R) by a custom LabVIEW software.

**Data analysis.** The analysis of the ultra-fast-force-clamp data was carried out by a custom MATLAB algorithm which detects catenin-actin interactions by identifying changes in the derivative of the position signal i.e., dumbbell velocity which drops to zero upon binding. Limits were imposed to ensure that false events were kept at a maximum of 1% of the total event number[10,30]. Step detection was performed as in Gardini et al. with small modifications[12]. Steps were detected from velocity changes due to proteins experiencing conformational changes or rapid detachment and reattachment to the filament. In this case, step length (>5 ms) and absolute amplitude (>5 nm) thresholds were set to limit detection of false steps due to thermal fluctuations, while step amplitude sign switch was allowed to account for both forward and backward steps. In order to distinguish if the steps were within an interaction or belonged to new

interactions, maximum time interval (<10 ms) and step size (<75 nm) were set. The average stiffness of the molecule (or molecules) bound to the actin filament during the interactions was evaluated from the position variance $\sigma_x^2$ during the bound events as $k = k_B T/\sigma_x^2$, where $k_B$ is the Boltzmann constant and $T$ is the absolute temperature. Here, we assume that (i) the dumbbell is rigid and (ii) the force applied on the trapped beads is constant, i.e. the stiffness of the traps is negligible under force-clamp. The stiffness of the molecules was evaluated as the average stiffness from the experiments reported in Figs. 1–3 in the range 7–16 pN force and separately computed for the single α-catenin homodimer, the single α-β-catenin heterodimer, and multiple α-β-catenin heterodimers. We expect that the approximation of rigid dumbbell leads to underestimate the α-catenin molecules stiffness, mostly because of the compliance of the links between actin and beads. However, since we previously measured the stiffness of the links between biotinylated actin and neutravidinated beads to range between 2 and 6 pN/nm under similar actin pretensioning[27], we evaluate the stiffness underestimation to be below 20%.

**Slip-bond model.** We fitted the lifetime vs force plot of a single α-β-catenin heterodimer with a Bell-bond model in which the lifetime τ decreases exponentially with force: $\tau = \tau_0^{\pm} \exp\left(-\frac{d_{\alpha\beta}^{\pm}F}{k_B T}\right)$, where $k_B$ is the Boltzmann constant, T the absolute temperature, F is the absolute value of the force, $\tau_0^{\pm}$ and $d_{\alpha\beta}^{\pm}$ are the unloaded lifetime and the distance parameter under positive ($^+$) and negative ($^-$) force respectively. The force sign is arbitrarily chosen. Fitting parameters are reported in the caption of Fig. 2b. OriginPro was used for data fitting.

**Two-step catch-bond model.** We divided the plot of the lifetime vs force for the α-catenin homodimer interaction with actin into a region of low force and high force to separate the lifetime peaks (Supplementary Fig. 1a). Assuming that within each force region α-catenin can transition between two structural states, we developed a two-step kinetic model to fit each lifetime peak. The model is based on two bound states, respectively weakly (W) and strongly (S) bound to actin, and one unbound state (U) (Supplementary Fig. 1b). α-catenin can switch between state W and S and vice-versa with transition rates $k_{WS}$ and $k_{SW}$. Detachment can occur from both states W and S with rates $k_{WU}$ and $k_{SU}$, respectively, with $k_{WU} \gg k_{SU}$. All rates depend exponentially on force $k_{ij} = k_{ij}^0 \exp\left(\frac{d_{ij}F}{k_B T}\right)$, with distance parameters $d_{WS}, d_{WU}, d_{SU} > 0$ and $d_{SW} < 0$ when $F > 0$. With $F < 0$ signs of distance parameters are changed. Therefore, if $k_{SW}^0 > k_{WS}^0$, α-catenin spends most of its time in state W at zero force. As the force increases, $k_{WS}$ increases and $k_{SW}$ decreases, so that, above a threshold force, state S becomes more populated than state W. The kinetic model can be solved following Nolting[34] to get the measured lifetime $\tau(F)$, which depends on the detachment rates $k_{WU}$ and $k_{SU}$ from both bound states W and S and on the transition rates $k_{WS}$ and $k_{SW}$ between them. The probability density function of the bound state ([W]+[S] = 1-[U]) is:

$$f(t) = -C_2\lambda_1\exp\left(-\lambda_1 t\right) - C_4\lambda_2\exp\left(-\lambda_2 t\right) \quad (2)$$

$$C_2 = \frac{k_{SW} + k_{WS} + k_{SU} - \lambda_1}{\lambda_1 - \lambda_2} \quad (3)$$

$$C_4 = -\frac{k_{SW} + k_{WS} + k_{SU} - \lambda_2}{\lambda_1 - \lambda_2} \quad (4)$$

$$\lambda_{1,2} = \frac{1}{2}\left(a \pm \sqrt{a^2 - 4b}\right) \quad (5)$$

$$a = k_{SW} + k_{WS} + k_{WU} + k_{SU} \quad (6)$$

$$b = k_{SW}k_{WU} + k_{WS}k_{SU} + k_{WU}k_{SU} \quad (7)$$

where $C_2$ and $C_4$ are calculated with initial conditions [U] = [S] = 0, [W] = 1. From the probability density function, we can calculate the average lifetime of the bound state:

$$\tau = \frac{k_{SW} + k_{WS} + k_{SU}}{k_{SW}k_{WU} + k_{WS}k_{SU} + k_{WU}k_{SU}} \quad (8)$$

that is used to fit experimental data (Fig. 1C and Supplementary Fig. 6). Fit parameters are represented graphically in Supplementary Figure 1c and in Supplementary Table 1. OriginPro was used for data fitting.

**Model of the interaction between α-catenin and a sliding actin filament under constant force.** We develop a simple mechano-kinetic model of the interaction between N α-catenin molecules and a single actin filament. We hypothesize that the α-catenin molecules interact independently with the actin filament with an attachment rate $k_{on} = 1/\tau_{off}$, with $\tau_{off}$ being the average lifetime of the unbound state, and a detachment rate $k_{off} = 1/\tau_{on}$, with $\tau_{on}$ being the average lifetime of the

bound state (see Supplementary Fig. 10). The duty ratio

$$r = \frac{\tau_{on}}{\tau_{on} + \tau_{off}} = \frac{k_{on}}{k_{on} + k_{off}} \quad (9)$$

represents the fraction of time a single α-catenin molecule spends attached to actin, as well as the average fraction of α-catenin molecules bound to actin. Therefore, the average number of attached molecules is $n = N \cdot r$ and $N_{\min} = 1/r$ is the minimum number of molecules to assure that, on average, at least one molecule is bound to actin[16]. $N_{\min}$, thus, represents the minimum number of molecules required to maintain continuous contact with the actin filament.

When a constant force $F$ is applied to the actin filament and no molecule is attached to it, the filament slides at constant velocity $v^*$ against viscous drag: $v^* = F/\gamma$, where γ is the viscous drag coefficient of the filament in the buffer solution or cytosol. When one or more α-catenin molecules attach to actin, the filament velocity drops to zero, as we observe in our experiments under constant force. The filament, thus, moves only during the fraction of time when α-catenin molecules are detached, and the sliding velocity is obtained as:

$$v = \frac{\tau_{off} \cdot v^*}{\tau_{on} + \tau_{off}} = (1 - r)\frac{F}{\gamma} \quad (10)$$

**Reporting summary**. Further information on research design is available in the Nature Research Reporting Summary linked to this article.

## Data availability
The data that support this study are available from the corresponding author upon reasonable request. The data generated in this study – duration and step size of single interactions that were used to generate lifetime plots (Figs. 1c, 2b, 3b, 4b, c, Supplementary Fig. 1a, 6), duty ratio plots (Fig. 5d–f), and step size distributions (Figs. 1g–j, 2c, d, 3c–f, Supplementary Fig. 3, 5, 8), as well as raw data of densitometry analysis in pull-down experiments (Supplementary Fig. 11 and 12) – are provided in the Source Data file. Source data are provided with this paper.

## Code availability
The software code used for this study is available on request from the corresponding authors.

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

## Acknowledgements
We thank Dr. Noboru Ishiyama for the pET28a vector containing mouse αE-catenin. This work was supported by the European Union's Horizon 2020 research and innovation program under grant agreement no 871124 Laserlab-Europe, by the Italian Ministry of University and Research (FIRB "Futuro in Ricerca" 2013 grant n. RBFR13V4M2), the University of Florence grant MURPASS, and by Ente Cassa di Risparmio di Firenze. A.V. Kashchuk was supported by the Human Frontier Science Program Cross-Disciplinary Fellowship LT008/2020-C.

## Author contributions
M.C. designed the research and supervised the experiments, C.A. expressed and purified α-catenin constructs, C.A. set up and performed optical trapping experiments, G.B. and A.V.K performed experiments on the oriented dumbbell, M.S. analyzed data, M.S., L.G. and M.C. developed the step detection analysis, G.B. performed pull-down experiments, C.A. and L.G performed flow-cell experiments, I.P. and P.B. setup the actin-tailed beads for the oriented dumbbell experiments, M.C., M.S., C.A. wrote the paper, M.S. prepared the figures, M.C., M.S., C.A., L.G., and F.S.P. revised the paper.

## Competing interests

The authors declare no competing interests.
