## [Peer Review File · Nature Communications]

α -catenin switches between a slip and an asymmetric catch bond with F-actin to cooperatively regulate cell junction fluidityReviewers' Comments:

Reviewer #1:

Remarks to the Author:

The authors investigated the force resistance of α - β -catenin heterodimer and α -catenin homodimer when they are engaged with a F-actin filament. The measurements were done using an ultra-fast laser tweezers. Main claims are 1) A single α - β -catenin heterodimer slips along F-actin in the direction of force, whereas multiple α - β -catenin heterodimers together with unidirectional force applied to F-actin lead to a switch to strong binding to F-actin exhibiting a catch bond kinetics. 2) An single α -catenin homodimer forms an asymmetric catch bond with F-actin. 3) Protein domain unfolding plays a critical role in the observed force-dependent lifetimes of kinetics of α - β -catenin heterodimers and α -catenin homodimer on F-actin. While the findings are interesting, major revisions with additional experiments will be needed to support the claims. Below are detailed comments.

The experimental design is similar to that developed by Buckley et al. in a previous study of the mechanical stability catenin/F-actin connection. The choice of investigating the mechanical stability of mammalian α - β -catenin heterodimer and α -catenin homodimers bound on F-actin, as well as the role of protein domain in α -catenin, confer novelties to the present study.

In Fig. 1a, the authors illustrates that in an α -catenin homodimer, only one α -catenin monomer binds to F-actin. Why do they think so? Is there any evidence? Why can't both α -catenins in the homodimer are engaged with F-actin? If both are engaged with F-actin, how will it affect the interpretation of the position steps observed at forces > 5 pN?

I don't quite understand Fig. 1c showing that a single α -catenin forms an asymmetric catch bond with F-actin. This asymmetric catch bond has to be dependent on the polarity of the F-actin. In the manuscript, the asymmetric catch bond is revealed by its dependence on the direction of the applied force. However, to make sense, a relation between the force direction and the polarity of the F-actin filament has to be established. In the experiment, F-actin was attached to microspheres via streptavidin-biotin linkage; therefore, it seems impossible to tell which force direction is toward the barbed end of the F-actin. Similarly, the authors should establish a relation between force direction and F-actin polarity for their measurement on α - β -catenin heterodimer.

The finding that a single α - β -catenin heterodimer does not bear force is surprising to me. It is reasonable to assume that the force-bearing property is mainly dependent on the C-terminal ABD of α -catenin. Therefore, one may expect that a single α - β -catenin heterodimer bound on F-actin and a single α -catenin heterodimer bound on F-actin using one of the α -catenin should have similar force-dependent lifetime on F-actin, which was not observed in their experiments. What could be a possible explanation? Is it possible that in an α -catenin homodimer, both α -catenin monomers bind F-actin, whereas in an α - β -catenin heterodimer, only an α -catenin monomer binds F-actin, which causes the difference?

For the α - β -catenin heterodimer, the authors observed extremely short lifetime with F-actin in the millisecond time scale. How do they know such short-lived signals are from α - β -catenin heterodimer? I can imagine that nonspecific friction between the F-actin and the bead may introduce short-lived signals. What are the controls to distinguish between α - β -catenin heterodimer mediated specific signals and friction mediated non-specific signals?

To test their hypothesis that multiple α - β -catenin heterodimers may form a more mechanically stable engagement with F-actin, the authors increased the concentration of α - β -catenin heterodimers by ten folds. However, there is a concern that at such elevated concentration, α -catenin homodimers may also be produced in a non-negligible concentration that could partly contribute to the observed signals. In fact, the reason why the Danio Rerio α -catenin was used in the previous study by Buckley et al. was to eliminate the potential presence of α -catenin homodimers that could complicate the

interpretation of cadherin-catenin complex/F-actin binding events.

The authors claimed that multiple α - β -catenin heterodimers form a catch bond with F-actin in a cooperative manner, since "a mere summation of multiple slip bonds would result again in a slip bond." This is a weak argument, because 1) they didn't rule out the possibility of the potential presence of α -catenin homodimers, 2) they didn't provide a possible mechanism on how multiple α - β -catenin heterodimers may cooperate to form such a catch-bond behavior.

In Fig. 4c, the authors illustrates that multiple α - β -catenin heterodimers bind F-actin simultaneously. Can they estimate the average number of the engaged α - β -catenin heterodimers on F-actin? In addition, if there are multiple simultaneously engaged α - β -catenin heterodimers, they should share the force. As a result, the domain unfolding force is expected to be much larger than 5 pN.

Reviewer #2:

Remarks to the Author:

This is an interesting study on an important problem that introduces new techniques of looking at a biological question. Please read below my thoughts on how the conclusions could be strengthened in my opinion, such as more information on the experimental setup.

Please elaborate on the physiological readout of the observed weak and rapid interactions and their biological implications

Including what does it mean for alpha-catenin interacting with itself in as a homodimer versus with beta-catenin

Line 322 - Please confirm that the recombinant protein is full-length

Line 333 – if GST was not cleaved as it seems, then what effects does the GST dimer have on this study?

Line 335 – likewise, what are the biotinylation effects on actin in this study?

Paragraph of line 364 – to which protein region does the bead attach and what are the steric constraints caused by this attachment?

Supplementary Figure 9a is missing controls, such as the proteins by themselves and pellet and supernatant fractions for each; also, while taking into account that the increasing amounts of alpha-catenin would show bands that are not visible at lower concentrations, the protein seems to degrade at higher concentrations, please comment

Reviewer #3:

Remarks to the Author:

The manuscript entitled " α -catenin switches between a slip and a cooperative catch bond with F-actin to regulate cell junction fluidity" by Arbore et al, investigates the force-dependent binding interaction between both α -catenin homodimers, and α - β -catenin heterodimers with F-actin using single molecule optical tweezers. By varying the direction of the applied force, the authors first explore the mechanical relationship between F-actin and a single α -catenin homodimer and report an asymmetric catch bond with F-actin. The authors proceed to investigate how the behavior of a single α - β -catenin heterodimer compares to the behavior of a cluster of α - β -catenin heterodimers, reporting that only the cluster of molecules is able to resist the force. Finally, the authors suggest that these finding lead to a phase

transition at the membrane-cytoskeleton interface.

Our current understanding of how mechanical force regulates the binding interactions of physiologically relevant molecules is limited. Furthermore, trying to understand how the directionality of force with respect to the orientation of the binding interaction is crucial in understanding many mechanically regulated biological processes. Therefore, this work is definitely relevant and is taking steps towards addressing key gaps in the knowledge of the field.

Despite the interesting topic, I have some concerns regarding the analysis / interpretation of the data, which impacts some of the key conclusions drawn by the authors. My primary concern is regarding the results demonstrating asymmetry in the force-dependent binding interactions, and I believe more controls are required to fully substantiate these claims. I have listed below a number of concerns, and I think that until these are addressed, the manuscript is not suitable for publication in Nature Communications.

1. A major finding of the manuscript is regarding the concept that the directionality of the force changes the binding interaction between α -catenin/ α - β -catenin and actin. The authors control the direction of the force, by changing the direction that the F-actin traps are moving with respect to the stationary catenin molecule(s). However, in order to understand how the direction of the force is altering the binding interaction, I believe it is critically important to understand the orientation of the molecules with respect to the force. As far as I can tell, the actin filament is biotinylated at both ends, and therefore the authors don't control the orientation of the actin in the trap. Therefore, the data will be a combination of both actin orientations (e.g sometimes moving the trap to the right will correspond towards moving towards the barbed end of actin, and other times it will correspond to the pointed end of actin). Therefore, I would expect that any asymmetry in the binding interaction would become smeared out due to the pooling of the data from different orientations.

The Dunn lab has developed an approach to ensure that bond asymmetry is not due to a systematic error of the optical traps, by rotating the traps 180 on the same molecule and seeing that the asymmetry swaps, therefore ensuring that the traps are not causing the changes. I believe that the authors should perform this experiment to ensure no result is originating from a trap error. Furthermore, Dunn lab has developed an assay to determine the barbed and pointed end of the actin filament and use this as the point of reference for the directionality of the force (Huang et al. Science, 2017. Owen et al. Biorxiv, 2020).

I believe if the authors want to report on the asymmetry of the binding interaction, it is fundamentally important that they determine the orientation of actin, and use this as the reference for the direction of the force and not simply use the direction of trap movement. This will provide an insight into the molecular mechanisms governing the binding interaction. Furthermore, this may help towards understanding the binding interaction that occurs in the native state of the cell.

2. The authors next compare the force dependent behavior of a single α - β -catenin heterodimer to the behavior of multiple α - β -catenin heterodimers with actin, and report that the monomer displays slip bond properties, compared to the catch bond behavior of the multi-protein binding. I am unsure of the steps taken by the authors to ensure single vs multiple protein functionalization of the silica sphere. In the abstract the authors go as far as to say that between 5 and 10 molecules are bound to the surface, but I am not able to find how the authors calculate this amount?

Furthermore, in the single protein trajectory of figure 1.e, it appears that there are 2 unfolding events (the step below the third arrow), but this trace should correspond to a single α -catenin homodimer. I am aware that there is large variability (and struggles!) in the success of surface functionalization for single molecule experiments. Typically, when trying to look at a single molecule, there are simple steps to exclude multibinding events from the analysis (on this note, the authors do not explain if they implement these steps in their analysis). However, I think it is not so straightforward to control the amount of multiple molecules. Given that this comparison is a major finding of the manuscript, I think the authors need to have a more vigorous approach (either through experimental controls or analysis approaches) towards determining the amount of molecules bound to the actin filament.

3. The manuscript title, in combination with the final sentence of the abstract stating “Our data reveals that α -catenin clustering together with intracellular tension engage a fluid-to-solid phase transition at the membrane cytoskeleton interface” gives the impression of strong evidence of a phase transition at cell junctions arising from the binding between catenin and actin. However, I am not convinced that this claim is sufficiently substantiated. This concept of phase transition is only touched upon in the final paragraph of the discussion and is not supported by any experimental evidence. Therefore, I believe that the authors need to either provide additional evidence (either through experiments or a deeper explanation of the model) of a phase transition arising from the slip-to-catch bond transition, or tone down this statement throughout the manuscript.

4. The authors validate some of their protein unfolding observations by comparing to the results from Yao et al. manuscript. However, I am not clear if the protein construct used by Yao is the same as the construct used here. Therefore, given that the authors know the number of amino acids of the protein and the magnitude of applied force, I would urge them to calculate the expected unfolding step size and use this as the primary method to validate the unfolding event.

We thank the reviewers for their appreciation of our work and for the insightful comments that allowed us to improve our manuscript. Here, we include a point-by-point response to the reviewers' comments. Moreover, in the manuscript text file and in the supplementary information we highlighted all changes in yellow.

REVIEWER COMMENTS

Reviewer #1 (Remarks to the Author):

The authors investigated the force resistance of α - β -catenin heterodimer and α -catenin homodimer when they are engaged with a F-actin filament. The measurements were done using an ultra-fast laser tweezers. Main claims are 1) A single α - β -catenin heterodimer slips along F-actin in the direction of force, whereas multiple α - β -catenin heterodimers together with unidirectional force applied to F-actin lead to a switch to strong binding to F-actin exhibiting a catch bond kinetics. 2) An single α -catenin homodimer forms an asymmetric catch bond with F-actin. 3) Protein domain unfolding plays a critical role in the observed force-dependent lifetimes of kinetics of α - β -catenin heterodimers and α -catenin homodimer on F-actin. While the findings are interesting, major revisions with additional experiments will be needed to support the claims. Below are detailed comments.

The experimental design is similar to that developed by Buckley et al. in a previous study of the mechanical stability catenin/F-actin connection. The choice of investigating the mechanical stability of mammalian α - β -catenin heterodimer and α -catenin homodimers bound on F-actin, as well as the role of protein domain in α -catenin, confer novelties to the present study.

1 - We thank the reviewer for the positive comments about the importance of the results that we report in our manuscript

In Fig. 1a, the authors illustrates that in an α -catenin homodimer, only one α -catenin monomer binds to F-actin. Why do they think so? Is there any evidence? Why can't both α -catenins in the homodimer are engaged with F-actin?

If both are engaged with F-actin, how will it affect the interpretation of the position steps observed at forces > 5 pN?

2 - Fig. 1a illustrates a possible interaction of a single α -catenin monomer with actin, but we do not hypothesize that this is the only way the α -catenin dimer interacts with F-actin. To the best of our knowledge, there is no structure of the α -catenin dimer in complex with F-actin. However, structural studies of the α -catenin homodimer do not exclude that both monomers might bind F-actin simultaneously [Rangarajan and Izard, Nature Struct Mol Biol, 2013]. We have specified this at

the end of the first section of the results (lines 154-157) and discussed in detail how it can affect the interpretation of the position steps in the Supplementary Discussion (section “Step Size Distribution”) that we added at the end of the Supplementary Information.

Briefly, although binding of both monomers to actin would open to the possibility that the position steps observed at force > 5 pN are the consequence of the unbinding of one of the two monomers, we conclude that this cannot be the primary cause for the main peak in the step distribution. In fact, the step size of the main peak does not change significantly with force (it is about 12-14 nm for 5 pN and 11 pN force, see Fig. 1g-j), whereas if the step was a consequence of the change in stiffness of the bond after the unbinding of one of the two monomers, it should change with force. On the other hand, if the step was due to a force-induced conformational change of the protein that did not modify substantially the protein stiffness, the step would be independent of the clamped force, as we observe. However, we cannot exclude that the binding of two monomers to actin might affect the measured step size distributions and, in particular, the larger peaks observed at higher forces (Fig. 1j). This is now discussed in detail in the supplementary discussion (section “Step Size Distribution”).

I don't quite understand Fig. 1c showing that a single α -catenin forms an asymmetric catch bond with F-actin. This asymmetric catch bond has to be dependent on the polarity of the F-actin. In the manuscript, the asymmetric catch bond is revealed by its dependence on the direction of the applied force. However, to make sense, a relation between the force direction and the polarity of the F-actin filament has to be established. In the experiment, F-actin was attached to microspheres via streptavidin-biotin linkage; therefore, it seems impossible to tell which force direction is toward the barbed end of the F-actin. Similarly, the authors should establish a relation between force direction and F-actin polarity for their measurement on α - β -catenin heterodimer.

3 - We thank the referee for this comment and we apologize for the lack of clarity in the manuscript. In Fig. 1-3, we reported lifetimes and step size distributions acquired using the same dumbbell to maintain the polarity of the filament and highlight the asymmetry in the force direction that otherwise would have been canceled out by mixing measurements performed with filaments oriented in random directions. We added sentences in the main text to clarify this point (lines 91-92, 208-210, 220-222). However, we agree with the referee that we cannot establish the polarity of the actin filament with the streptavidin-biotin link used in these experiments. We struggled to find a way to determine the actin polarity in our experiments, first by trying dual labelling of the actin filament [Bryant, PNAS, 2007]. Unfortunately, this method did not work for technical difficulties related to the low efficiency of double-labelling and fast bleaching of the chromophores during trapping of the actin filament floating in solution. Finally, in collaboration with Dr. Pasquale Bianco and Irene Pertici, we succeeded in forming oriented dumbbells by producing actin-tailed beads in which the actin +end was specifically

bound to a gelsolin-coated bead [Pertici et al, IJMS, 2020]. The actin filament was also biotinylated so that the actin -end could bind to streptavidin-coated beads. The actin-tailed beads were not fluorescent by their own, but the attached rhodamine-phalloidin labelled filaments gave a high signal when excited at 532 nm, whereas the streptavidinated beads were coated with streptavidin-alexa 647, excited at 635 nm thanks to a second diode laser added to the setup. In this way we could build oriented dumbbells with known polarity (see methods, lines 435-445 and 471-483). The experiments performed on single α -catenin homodimers and multiple α - β -catenin heterodimers both showed a strong asymmetry. The lifetime was much longer when the force was applied towards the actin -end. We added a section and a figure in the manuscript where we describe these experiments and the associated results (lines 247-269 and Fig. 4). We discuss the biological relevance of this result in the discussion, lines 333-341.

The finding that a single α - β -catenin heterodimer does not bear force is surprising to me. It is reasonable to assume that the force-bearing property is mainly dependent on the C-terminal ABD of α -catenin. Therefore, one may expect that a single α - β -catenin heterodimer bound on F-actin and a single α -catenin heterodimer bound on F-actin using one of the α -catenin should have similar force-dependent lifetime on F-actin, which was not observed in their experiments. What could be a possible explanation? Is it possible that in an α -catenin homodimer, both α -catenin monomers bind F-actin, whereas in an α - β -catenin heterodimer, only an α -catenin monomer binds F-actin, which causes the difference?

4 – Yes, it is possible and it is what our data suggest, but other possibilities exists. We now discuss this point in the Supplementary Discussion, section “Cooperative Binding”, and replicate it below for the referee’s convenience.

It’s well established that the interaction between α - β -catenin heterodimers and actin is much weaker than between α -catenin homodimers and actin [Drees et al., Cell, 2005]. Different explanations have been proposed previously (i) the N- and C-terminal domains of α -catenin are allosterically coupled and binding to β -catenin on the N-terminal domain might alter the C-terminal domain ability to bind to actin [Drees et al., Cell, 2005]. (ii) Structural studies [Rangarajan and Izard, Nature Struct Mol Biol, 2013] indicate that β -catenin might sterically hinder F-actin binding by the α -catenin binding domain, which could be at the basis of the different F-actin binding between the homodimer and heterodimer. (iii) α -catenin ABD binding to actin is accompanied by a conformational change in the actin protomer that affects the filament structure. This alteration of the filament structure can be at the base of a cooperative binding mechanism that reinforces the link between an α -catenin homodimer and actin, when two α -catenin molecules interact, compared to a single α -catenin monomer [Hansen et al, Mol Biol Cell, 2013]. We underline this mechanism in the manuscript discussion (lines 302-305) because our results indicate that a cooperative mechanism is at the basis of the bond reinforcement. The new analysis of the molecule stiffness during the interaction with actin reinforces this interpretation (see reply n°9 below). However,

the identification of the structural features that are at the basis of the different kinetics of α -catenin homodimers and heterodimers is out of the scope of our article, and further studies would be required to clearly assess this point.

For the α - β -catenin heterodimer, the authors observed extremely short lifetime with F-actin in the millisecond time scale. How do they know such short-lived signals are from α - β -catenin heterodimer? I can imagine that nonspecific friction between the F-actin and the bead may introduce short-lived signals. What are the controls to distinguish between α - β -catenin heterodimer mediated specific signals and friction mediated non-specific signals?

5 - We made many control experiments to rule out non-specific interactions, which are one of the well-known issues in this kind of single molecule experiments. In experiments on α - β -catenin heterodimers, α -catenin was attached on the coverslip surface on top of a nitrocellulose-coated surface saturated with β -catenin, followed by BSA (see methods). We made several control slides in which the coverslip surface was coated as described above but in the absence of α -catenin and looked for non-specific interactions on several tens of beads in each slide. We could very rarely (less than one bead per slide) find non-specific interactions with this control surface. Moreover, non-specific interactions were very different from the interactions observed in the presence of α -catenin, showing few short interactions when the dumbbell was close to the coverslip surface and the actin filament was pushing on the bead (as detected from the change in the position signal) and disappeared when the dumbbell was moved slightly farther from the coverslip surface. On the other hand, in the presence of α -catenin at single molecule concentration, we observed interactions in one every 4 beads on average, the interactions were much longer at low forces (tens of milliseconds) and the number of interactions increased with force. A single molecule was able to produce as much as several tens of thousands of interactions. The interactions were observed also when the actin filament was not pushing on the bead. This behavior was never observed in the absence of α -catenin. We inserted a section in the supplementary discussion named "non-specific interactions" to describe in detail the control experiments for non-specific interactions and a short description in the methods (lines 464-467).

To test their hypothesis that multiple α - β -catenin heterodimers may form a more mechanically stable engagement with F-actin, the authors increased the concentration of α - β -catenin heterodimers by ten folds. However, there is a concern that at such elevated concentration, α -catenin homodimers may also be produced in a non-negligible concentration that could partly contribute to the observed signals. In fact, the reason why the Danio Rerio α -catenin was used in the previous study by Buckley et al. was to eliminate the potential presence of α -catenin homodimers that could complicate the interpretation of cadherin-catenin complex/F-actin binding events.

6 - Before our experiments, we separated dimeric from monomeric α -catenin by using size exclusion chromatography. This procedure assures that less than 1% of the catenin was in a dimeric form in the experiments with α - β -catenin heterodimers (see supplementary Fig. 9). The concentration of the monomeric catenin that we used in the experiments at single molecule concentration was about 1 μ g/ml (10 nM); in the ones at “high” concentration, catenin concentration was 10 μ g/ml (100 nM). Since the dissociation constant of the α -catenin homodimer is 25 μ M, at equilibrium about 0.04% and 0.4% would be dimeric at the single-molecule and high concentration, respectively. Moreover, Pokutta et al. showed that the α -catenin homodimer does not bind to β -catenin even after overnight incubation [Pokutta et al. *J. Biol. Chem.* 289, 13589-601, 2014]. Therefore, the few % contamination of dimeric catenin was most likely washed away after few minutes of incubation in the sample chamber (see methods). We avoided to use the *Danio Rerio* α -catenin because it has been shown that this catenin has a significantly different kinetics from mammalian catenin [Miller et al., *J. Biol. Chem.*, 288, 22324-32, 2013]. In particular, the authors showed that the interaction between *Danio Rerio* α -catenin complexed to β -catenin and actin is much stronger than for mammalian α -catenin. In fact, different from mammalian α -catenin, *Danio Rerio* α -catenin complexed to β -catenin co-precipitated with actin in a cosedimentation assay. We have added a sentence in the methods about the concerns on the possible artifactual contributions raised by the referee (lines 467-470) and reported these in the supplementary discussion (section “Dimeric vs monomeric α -catenin”).

The authors claimed that multiple α - β -catenin heterodimers form a catch bond with F-actin in a cooperative manner, since “a mere summation of multiple slip bonds would result again in a slip bond.” This is a weak argument, because 1) they didn’t rule out the possibility of the potential presence of α -catenin homodimers,

7 - As discussed in the previous reply to the referee, the formation of α -catenin homodimers that bind to the β -catenin coated surface on the coverslip is very unlikely.

2) they didn’t provide a possible mechanism on how multiple α - β -catenin heterodimers may cooperate to form such a catch-bond behavior.

8 - The cooperativity of the interaction between α -catenin and actin is well established [Hansen et al, *Mol Biol Cell*, 2013]. We already discussed this in our reply n.4 to the referee. In the paper discussion (lines 302-305) we provide a possible mechanism on how multiple α - β -catenin heterodimers may cooperate to form a catch-bond: “Previous studies indicate that α -catenin binds cooperatively to F-actin by altering the actin filament conformation²⁰, which might be the molecular mechanism at the base of the change in the load-dependence from single to multiple α - β -catenin heterodimers that we observed.”. We now discuss

this point more thoroughly in the “cooperative binding” section of the supplementary discussion.

In Fig. 4c, the authors illustrates that multiple α - β -catenin heterodimers bind F-actin simultaneously. Can they estimate the average number of the engaged α - β -catenin heterodimers on F-actin? In addition, if there are multiple simultaneously engaged α - β -catenin heterodimers, they should share the force. As a result, the domain unfolding force is expected to be much larger than 5 pN.

9 - From the duty ratio of the interaction between a single α - β -catenin and actin (0.1-0.2), we calculated that 5 to 10 catenins must be available for binding to actin to get one molecule bound to actin on average. Under this condition, occasionally, two catenins bind simultaneously to actin and can engage a cooperative action. By increasing the catenin concentration 10 times compared to the single molecule concentration, we observed a catch bond behavior with an increase in lifetime with force, as for the homodimer, with lifetimes comparable to the homodimer itself (Fig. 1c and 3b). This result, together with the fact that the unfolding force does not change significantly compared to the homodimer, is a good indication that the number of molecules that bind simultaneously to actin cannot be much greater than two.

Following the referee’s suggestion, we tried to estimate the average number of α -catenins bound to actin. To this end, we measured the average stiffness of the molecule during the interaction with actin from the position noise of the dumbbell (see methods, lines 514-527). The single α - β -catenin heterodimer gave an average stiffness of 0.263 ± 0.006 pN/nm (error s.e.m), about half of the α -catenin homodimer (0.46 ± 0.03 pN/nm). Assuming that the stiffness of an α -catenin monomer in the homodimer and in the α - β -catenin heterodimer is the same, this result suggests that both α -catenins in the homodimer participate to the interaction with actin for most of the time (~ 87% from the ratio of the homodimer stiffness to double the heterodimer stiffness). In the experiments with multiple α - β -catenin heterodimers, the average stiffness was in between the single heterodimer and the single homodimer (0.336 ± 0.007 pN/nm), suggesting that simultaneous binding of two α -catenin molecules occurred for about 64% of the time in these experiments. This result reinforces our conclusion that the catch bond behavior is a consequence of the cooperative binding of multiple α -catenin molecules to actin. We now report these results in the discussion (lines 309-322). We modified the cartoon of Fig. 3, 4 and 5 to indicate that about two molecules interact simultaneously with actin under our experimental conditions.

Reviewer #2 (Remarks to the Author):

This is an interesting study on an important problem that introduces new techniques of looking at a biological question. Please read below my thoughts on

how the conclusions could be strengthened in my opinion, such as more information on the experimental setup.

1 - We thank the reviewer for the positive comments about the importance of the results that we report in our manuscript

Please elaborate on the physiological readout of the observed weak and rapid interactions and their biological implications Including what does it mean for alpha-catenin interacting with itself in as a homodimer versus with beta-catenin

2 - We have extended the discussion and added a supplementary discussion in the supplementary information text to argue about the possible mechanisms behind the switch between the weak and rapid interactions of the α - β -catenin heterodimer compared to the α -catenin homodimer or multiple α - β -catenin heterodimers. In particular, in the section “cooperative binding” of the supplementary discussion we consider the possible structural changes induced by the interaction of α -catenin with itself or with β -catenin.

Line 322 - Please confirm that the recombinant protein is full-length

3 - We now specify in the methods that the α E-catenin protein used in the experiments is full-length (line 386).

Line 333 – if GST was not cleaved as it seems, then what effects does the GST dimer have on this study?

4 - In our sample chamber, we first incubate β -catenin at high concentration to saturate the coverslip surface. We then incubate α -catenin at lower concentration to regulate the protein concentration on the surface (see methods). For this reason, the possible dimerization of β -catenin through the GST tag is not expected to influence the surface distribution of α -catenin or change the α -catenin dimeric or oligomeric state.

Line 335 – likewise, what are the biotinylation effects on actin in this study?

5 – We made biotinylated actin filaments by mixing biotinylated with non-biotinylated G-actin from Cytoskeleton Inc. The biotinylation efficiency and/or the fraction of actin monomers in which a biotin tag is exposed in the polymerized filament must be quite low. In fact, we had to mix biotinylated and non-biotinylated G-actin in a molar ratio ranging from 1:1 to 2:1 in order to get actin filaments that attached to the trapped streptavidinated beads. By probing the filaments with a streptavidinated bead, we found few binding sites on several micron long actin filaments; some filaments did not bind to streptavidinated beads at all. With such

a sparse biotinylation we expect that the influence on α -catenin binding would be negligible. Biotinylated actin filaments are routinely used in this kind of single molecule experiments (including the experiments on α -catenin by the Dunn's group [Buckley et al., Science, 2014]) and it has been generally found that actin biotinylation does not influence the binding of proteins to the filament significantly.

Paragraph of line 364 – to which protein region does the bead attach and what are the steric constraints caused by this attachment?

6 - We attach catenin non-specifically onto silica beads coated with a nitrocellulose substrate. Therefore, it is difficult to predict which protein region binds to the bead and we expect a large molecule-to-molecule variability. In this kind of experiments in which non-specific binding is used, the molecules that can bind to actin are “naturally selected” as the ones in which the actin binding region is properly oriented, far from the bead surface, and with enough protein mobility to allow readily binding to actin.

Supplementary Figure 9a is missing controls, such as the proteins by themselves and pellet and supernatant fractions for each; also, while taking into account that the increasing amounts of alpha-catenin would show bands that are not visible at lower concentrations, the protein seems to degrade at higher concentrations, please comment

As observed by the referee, when a large quantity of α -catenin pellet is loaded in the gel after the cosedimentation assay, it is possible to observe tiny bands of lower molecular weight, suggesting possible protein degradation. In particular, there's a brighter band just below the main α -catenin band, that we quantify from densitometry analysis to be around 10% of the main α -catenin band. However, this band is not visible in the α -catenin SDS-page gels that we did after purification (we added a supplementary figure, now named supplementary figure 9, with this gel). It is therefore possible that, in the pellet of the cosedimentation assay, we collect degraded protein that aggregate and precipitate with actin. Regarding the old Supplementary Fig. 9, now named supplementary Fig. 11, since we mixed known quantities of α -catenin and actin, we collected the pellet only, which was sufficient to measure the relative fraction of α -catenin bound to F-actin and the dissociation constant.

Reviewer #3 (Remarks to the Author):

The manuscript entitled “ α -catenin switches between a slip and a cooperative catch bond with F-actin to regulate cell junction fluidity” by Arbore et al,

investigates the force-dependent binding interaction between both α -catenin homodimers, and α - β -catenin heterodimers with F-actin using single molecule optical tweezers. By varying the direction of the applied force, the authors first explore the mechanical relationship between F-actin and a single α -catenin homodimer and report an asymmetric catch bond with F-actin. The authors proceed to investigate how the behavior of a single α - β -catenin heterodimer compares to the behavior of a cluster of α - β -catenin heterodimers, reporting that only the cluster of molecules is able to resist the force. Finally, the authors suggest that these findings lead to a phase transition at the membrane-cytoskeleton interface.

Our current understanding of how mechanical force regulates the binding interactions of physiologically relevant molecules is limited. Furthermore, trying to understand how the directionality of force with respect to the orientation of the binding interaction is crucial in understanding many mechanically regulated biological processes. Therefore, this work is definitely relevant and is taking steps towards addressing key gaps in the knowledge of the field. Despite the interesting topic, I have some concerns regarding the analysis / interpretation of the data, which impacts some of the key conclusions drawn by the authors. My primary concern is regarding the results demonstrating asymmetry in the force-dependent binding interactions, and I believe more controls are required to fully substantiate these claims. I have listed below a number of concerns, and I think that until these are addressed, the manuscript is not suitable for publication in Nature Communications.

1 - We thank the reviewer for the positive comments about the importance of the results that we report in our manuscript

1. A major finding of the manuscript is regarding the concept that the directionality of the force changes the binding interaction between α -catenin/ α - β -catenin and actin. The authors control the direction of the force, by changing the direction that the F-actin traps are moving with respect to the stationary catenin molecule(s). However, in order to understand how the direction of the force is altering the binding interaction, I believe it is critically important to understand the orientation of the molecules with respect to the force. As far as I can tell, the actin filament is biotinylated at both ends, and therefore the authors don't control the orientation of the actin in the trap. Therefore, the data will be a combination of both actin orientations (e.g. sometimes moving the trap to the right will correspond towards moving towards the barbed end of actin, and other times it will correspond to the pointed end of actin). Therefore, I would expect that any asymmetry in the binding interaction would become smeared out due to the pooling of the data from different orientations.

2- We thank the referee (and referee#1 who made a similar comment) and we apologize for the lack of clarity in the manuscript. In Fig. 1-3, we report lifetimes and step size distributions acquired using the same dumbbell to maintain the polarity of the filament and highlight the asymmetry in the force direction that

otherwise would have been canceled out by mixing measurements performed with filaments oriented in random directions. We added sentences in the main text to clarify this point (lines 91-92, 208-210, 220-222). However, we agree with the referee that we cannot establish the polarity of the actin filament with the streptavidin-biotin link used in these experiments (see next point).

The Dunn lab has developed an approach to ensure that bond asymmetry is not due to a systematic error of the optical traps, by rotating the traps 180 on the same molecule and seeing that the asymmetry swaps, therefore ensuring that the traps are not causing the changes. I believe that the authors should perform this experiment to ensure no result is originating from a trap error. Furthermore, Dunn lab has developed an assay to determine the barbed and pointed end of the actin filament and use this as the point of reference for the directionality of the force (Huang et al. *Science*, 2017. Owen et al. *Biorxiv*, 2020).

I believe if the authors want to report on the asymmetry of the binding interaction, it is fundamentally important that they determine the orientation of actin, and use this as the reference for the direction of the force and not simple use the direction of trap movement. This will provide an insight into the molecular mechanisms governing the binding interaction. Furthermore, this may help towards understanding the binding interaction that occurs in the native state of the cell.

As suggested by the referee and by referee #1, we struggled to find a way to determine the actin polarity. We first tried dual labelling of the actin filament [Bryant, *PNAS*, 2007] but, unfortunately, this method did not work for technical difficulties related to the low efficiency of double-labelling and fast bleaching of the chromophores during trapping of the actin filament floating in solution. Finally, in collaboration with Dr. Pasquale Bianco and Irene Pertici, we succeeded in forming oriented dumbbells by producing actin-tailed beads in which the actin +end was specifically bound to a gelsolin-coated bead [Pertici et al, *IJMS*, 2020]. The actin filament was also biotinylated so that the actin -end could bind to streptavidin-coated beads. The actin-tailed beads were not fluorescent by their own, but the attached rhodamine-phalloidin labelled filaments gave a high signal when excited at 532 nm, whereas the streptavidinated beads were coated with streptavidin-alexa 647, excited at 635 nm thanks to a second diode laser added to the setup. In this way we could build oriented dumbbells with known polarity (see methods, lines 435-445 and 471-483). The experiments performed on single α -catenin homodimers and multiple α - β -catenin heterodimers both showed a strong asymmetry. The lifetime was much longer when the force was applied towards the actin -end. We added a section and a figure in the manuscript where we describe these experiments and the associated results (lines 247-269 and Fig. 4). We discuss the biological relevance of this result in the discussion, lines 333-341. Data reported in Fig. 4 are from experiments performed with different dumbbells. The dumbbell orientation with respect to the two traps was changed from one experiment to another to exclude that trap asymmetries might influence the results.

2. The authors next compare the force dependent behavior of a single α - β -catenin heterodimer to the behavior of multiple α - β -catenin heterodimers with actin, and report that the monomer displays slip bond properties, compared to the catch bond behavior of the multi-protein binding. I am unsure of the steps taken by the authors to ensure single vs multiple protein functionalization of the silica sphere. In the abstract the authors go as far as to say that between 5 and 10 molecules are bound to the surface, but I am not able to find how the authors calculate this amount?

Furthermore, in the single protein trajectory of figure 1.e, it appears that there are 2 unfolding events (the step below the third arrow), but this trace should correspond to a single α -catenin homodimer. I am aware that there is large variability (and struggles!) in the success of surface functionalization for single molecule experiments. Typically, when trying to look at a single molecule, there are simple steps to exclude multibinding events from the analysis (on this note, the authors do not explain if they implement these steps in their analysis). However, I think it is not so straightforward to control the amount of multiple molecules. Given that this comparison is a major finding of the manuscript, I think the authors need to have a more vigorous approach (either through experimental controls or analysis approaches) towards determining the amount of molecules bound to the actin filament.

Regarding the experiments with single molecules, we determined the single molecule concentration in our experiments in the usual way it is done in this kind of experiments by most research groups. In this procedure, we scan silica beads on the surface by moving the actin dumbbell along the bead and look for interactions in the position signal. At high α -catenin concentration we find several positions on the bead showing interactions. We progressively decrease the α -catenin concentration until approximately 1 in 4 silica beads show interactions with the actin filament in one specific position on the bead, providing evidence that the large majority of the beads contains at most a single catenin molecule (see methods, lines 461-463).

Under the single molecule concentration condition obtained as described above, we observed single steps and occasionally multiple steps, especially at large force (see supplementary figure 2c). Since only one molecule is interacting, the multiple steps could be caused by additional unfolding steps, which has been observed previously [Yao et al., Nat Comm, 2014], fast unbind and rebinding of catenin, or a contribution from the second catenin monomer in the dimeric α -catenin. We explicitly mention the observation of multiple steps in the revised paper (line 130) and discuss the possible causes (lines 151-157). We also discuss the possible contribution of the second catenin monomer in the dimeric α -catenin to the observed steps in the supplementary discussion (section "step size distribution"). Regarding the experiments with multiple α - β -catenin heterodimers, since the measured duty ratio of the single α - β -catenin heterodimer is between 0.1-0.2, we calculated that 5 to 10 catenins must be available for binding to actin to get one

molecule bound to actin on average. Under this condition, occasionally, two catenins bind simultaneously to actin and can engage a cooperative action. Therefore, our estimation that 5 to 10 catenins must be present to engage a cooperative action derives from the measurement of the duty ratio of the single α - β -catenin heterodimer (see methods, lines 570-583).

In the experiments in which we increased the α - β -catenin heterodimer concentration 10 times with respect to the single molecule concentration, we observed a catch bond behavior with an increase in lifetime with force, as for the α -catenin homodimer. The change from sleep to catch bond indicates that a cooperative action between multiple molecules occurred in these experiments, i.e. more than one molecule was interacting with the actin filament. Moreover, the lifetime and the unfolding force were comparable to the ones of the homodimer, suggesting that the number of molecules interacting simultaneously with actin could not be much greater than two.

Following the referee's suggestion, we tried to estimate the average number of α -catenins bound to actin. To this end, we measured the average stiffness of the molecule during the interaction with actin from the position noise of the dumbbell (see methods, lines 514-527). The single α - β -catenin heterodimer gave an average stiffness of 0.263 ± 0.006 pN/nm (error s.e.m), about half of the α -catenin homodimer (0.46 ± 0.03 pN/nm). Assuming that the stiffness of an α -catenin monomer in the homodimer and in the α - β -catenin heterodimer is the same, this result suggests that both α -catenins in the homodimer participate to the interaction with actin for most of the time ($\sim 87\%$ from the ratio of the homodimer stiffness to double the heterodimer stiffness). In the experiments with multiple α - β -catenin heterodimers, the average stiffness was in between the single heterodimer and the single homodimer (0.336 ± 0.007 pN/nm), suggesting that simultaneous binding of two α -catenin molecules occurred for about 64% of the time in these experiments.

These results reinforce our conclusion that the catch bond behavior is a consequence of the cooperative binding of multiple α -catenin molecules to actin and that, in the experiments with an α -catenin homodimer and with multiple α - β -catenin heterodimer, two α -catenin molecules cooperate in the interaction. We now report these results in the discussion (lines 309-322). We modified the cartoon of Fig. 3, 4 and 5 to indicate that about two molecules interact simultaneously with actin under our experimental conditions.

3. The manuscript title, in combination with the final sentence of the abstract stating "Our data reveals that α -catenin clustering together with intracellular tension engage a fluid-to-solid phase transition at the membrane cytoskeleton interface" gives the impression of strong evidence of a phase transition at cell junctions arising from the binding between catenin and actin. However, I am not convinced that this claim is sufficiently substantiated. This concept of phase transition is only touched upon in the final paragraph of the discussion and is not supported by any experimental evidence. Therefore, I believe that the authors

need to either provide additional evidence (either through experiments or a deeper explanation of the model) of a phase transition arising from the slip-to-catch bond transition, or tone down this statement throughout the manuscript.

Our data shows that the duty ratio is roughly 0.1-0.2, independent of force for the single heterodimer, while it increases close to 1 for multiple heterodimers under force. The simple model that we developed shows that the force-velocity relationship depends on the duty ratio and a constant duty ratio corresponds to an actin sliding velocity proportional to force. This means that catenin applies a drag force to actin and the actin-catenin link behaves as fluid. On the other hand, when the duty ratio approaches 1, the velocity drops close to zero and the filament stalls, as it occurs in elastic linkages. We moved the model from the supplementary methods to the methods (lines 568-590) and explained it more in detail in the discussion (lines 360-380). Although the model assumptions are reasonable and our data demonstrate the change in the response of the catenin-actin linkage, we agree with the referee that our results do not demonstrate what actually happens in cells, where the interactions are much more complex. As suggested by the referee, we toned down our claims in the abstract changing “reveals” with “suggest”.

4. The authors validate some of their protein unfolding observations by comparing to the results from Yao et al. manuscript. However, I am not clear if the protein construct used by Yao is the same as the construct used here. Therefore, given that the authors know the number of amino acids of the protein and the magnitude of applied force, I would urge them to calculate the expected unfolding step size and use this as the primary method to validate the unfolding event.

Yao et al. used the central domain of α -catenin. Although this is a fraction of the full length used in our work, structural and sequence analysis indicate that unfolding of α -catenin occurs in the central domain region [Maiden and Hardin, *J Cell Biol* **195**, 543–552 (2011)]. This means, that, in principle we should observe the same unfolding step size for the full length and central domain. In the discussion of their article, Yao et al. estimated the expected unfolding step size from the number of amino acids of the protein and the magnitude of the applied force to be about 15 nm. We now mention this in the manuscript discussion (lines 327-329).

Although the unfolding step size estimated from α -catenin structure and the values measured by Yao et al and us are approximatively the same, we would like to underline that it is difficult to make quantitative comparison of structural data and measurements performed with very different experimental configurations. In particular, the geometry of attachment and force application of our experiment differs from the one used by Yao et al. Yao et al. attachment to the surface and magnetic bead are both specific and force is applied vertically. In our experiments, catenin binds to actin and non-specifically to the coverslip-bound bead, while force is applied horizontally in the direction parallel to the actin filament. Moreover, we

observe unfolding using a dimeric protein or multiple heterodimers, which further complicates quantitative comparisons.

Reviewers' Comments:

Reviewer #1:

Remarks to the Author:

The authors have made substantial efforts to revise the manuscript and have provided important new experimental data to support their main conclusions. They have also satisfactorily responded to most of my comments. I recommend publication of the work.

Reviewer #2:

Remarks to the Author:

This is an interesting study on an important problem that introduces new techniques of looking at a biological question. Two technical points that were not addressed in the revised version of the manuscript but to me they are highly significant in terms of being able to draw conclusions and therefore significantly lower my enthusiasm:

1. To me, it does not make sense to leave the GST tag instead of cleaving it as the GST introduces lots of artifacts in terms non-specific interactions with GST but more importantly because GST is a dimer.

2. This was not addressed: the original Supplementary Figure 9a (now unmodified despite my comments Supplementary Figure 11a) does not have controls, such as the alpha-catenin and actin spun by themselves, the supernatant fractions are missing, and the molecular weight markers are missing

Reviewer #3:

Remarks to the Author:

I think that the additional experiments, analysis and clarification of the text has greatly improved the manuscript. The authors have addressed all of my previous concerns, and I support the publication of the manuscript.

We thank the reviewers for their appreciation of our work and for the insightful comments that allowed us to significantly improve our manuscript. Here, we include a point-by-point response to the reviewers' comments. Moreover, in the manuscript text file and in the supplementary information we highlighted all changes in yellow.

REVIEWER COMMENTS

Reviewer #1 (Remarks to the Author):

The authors have made substantial efforts to revise the manuscript and have provided important new experimental data to support their main conclusions. They have also satisfactorily responded to most of my comments. I recommend publication of the work.

We thank the reviewer for the positive comments and the recommendation to publish our manuscript

Reviewer #2 (Remarks to the Author):

This is an interesting study on an important problem that introduces new techniques of looking at a biological question. Two technical points that were not addressed in the revised version of the manuscript but to me they are highly significant in terms of being able to draw conclusions and therefore significantly lower my enthusiasm:

1. To me, it does not make sense to leave the GST tag instead of cleaving it as the GST introduces lots of artifacts in terms non-specific interactions with GST but more importantly because GST is a dimer.

4 –The GST tagged β -catenin was bought from Sigma Aldrich (SRP5172) and it was not feasible to remove the GST tag using proteolysis with thrombin because of the small quantity of protein available and the protein loss after this procedure. We agree with the referee that the presence of the GST tag might in principle influence the interaction of β -catenin with α -catenin and actin. However, previous studies reported that the GST tag does not affect significantly these interactions [Rimm et al. PNAS pp. 8813-8817, 1995] and we have direct evidence that the GST tag does not affect significantly our results, as detailed below.

Regarding “non-specific interactions with GST” mentioned by the referee, as reported in the previous version of the manuscript, we checked non-specific

interactions between the GST-tagged β -catenin and actin in both the flow cell assay and single molecule experiments. Using the flow cell assay, we did not observe binding of actin filaments on the coverslip surface coated with GST-tagged β -catenin (see Supplementary Methods, section “flow cell assay”, lines 237-242 and Supplementary Fig. 14d). In single molecule experiments, we made several control coverslips coated with the GST-tagged β -catenin and very rarely observed non-specific interactions with an actin filament, similarly to what is observed in the absence of the GST-tagged β -catenin. These control experiments are described in the methods (lines 490-493) and, more in detail, in the supplementary discussion (section “non-specific interactions”, lines 320-338).

To evaluate whether the presence of the GST tag would affect the interaction between β -catenin and α -catenin, we performed new experiments, reported in the revised manuscript (see methods, lines 437-453 and supplementary materials lines 176-208). In particular, we made pull-down experiments in which we bound GST- β -catenin to microbeads and made them react with α -catenin at growing concentrations. From these experiments, we measured a dissociation constant of about 34 nM for the α -catenin- β -catenin interaction, in good agreement with previous values reported in literature [Koslov et al. JBC 272, p27301–27306, 1997; Pokutta et al. JBC 289, p13589-13601, 2014]. This result indicates that the interaction between α -catenin and β -catenin is not affected significantly by the presence of the GST tag.

Regarding the influence of the GST dimer in our experiments, we used β -catenin to coat the coverslip surface at saturating concentration and we bound α -catenin on top of this β -catenin carpet at lower concentration. Under this condition, we expect to find β -catenin molecules tightly packed on the coverslip surface regardless of their dimeric or monomeric state and α -catenin distribution on the surface dictated mostly by α -catenin concentration, not by β -catenin dimeric or monomeric state.

In support to this argument, we have strong experimental evidence that in the experiments at low α -catenin concentration we observe interactions with single α - β -catenin heterodimers. In fact, the 5.5 nm periodicity that we observe in our position data (Fig. 2d-f) can be observed only if one α -catenin molecule is interacting with actin, whereas α -catenin dimers or multiple molecules positioned randomly on the surface would average out this precise distribution. Therefore, it is very unlikely that single molecule experiments are affected by the presence of GST dimers.

At higher α -catenin concentration, it remains possible that two adjacent α -catenin molecules might be bound to two β -catenin dimerized through the GST tag, which might possibly be arranged differently from two adjacent α - β -catenin heterodimers that are not dimerized through the GST tag. It is always difficult to predict or measure how proteins are distributed and oriented in single molecule trapping experiments and the influence that this can have on the measured kinetics, which is a well-known limitation of the experimental method. The present experimental arrangement does not allow us to control how α - β -catenin heterodimers are distributed on the coverslip surface and if and how particular

arrangements of the molecules might affect their cooperative behavior. This is out of the scope of the present work and further studies would be needed to assess this point. We have however tried to clearly highlight the experimental conditions under which the experiments were performed to allow high reproducibility of the results. We have improved the revised manuscript in this respect by highlighting that we used GST-tagged β -catenin in the main text (lines 210-214) and we added a section in the supplementary discussion to describe the arguments reported in this response to the referee (section “Possible effect of the GST tag in β -catenin”, lines 353-399).

2. This was not addressed: the original Supplementary Figure 9a (now unmodified despite my comments Supplementary Figure 11a) does not have controls, such as the α -catenin and actin spun by themselves, the supernatant fractions are missing, and the molecular weight markers are missing

Following the referee request, we performed a new co-sedimentation assay, collecting also supernatant fractions, and running gels including purified α -catenin and actin, α -catenin and actin spun by themselves, and molecular weight markers. These new experiments are described in Supplementary Methods (section “ α -catenin-actin pull-down assay”, lines 156-172). The corresponding gels are shown in Supplementary Figure 11 together with the densitometry analysis, which confirms our previous measurement of the α -catenin-actin dissociation constant and values reported in literature within the uncertainties of the experimental method.

Reviewer #3 (Remarks to the Author):

I think that the additional experiments, analysis and clarification of the text has greatly improved the manuscript. The authors have addressed all of my previous concerns, and I support the publication of the manuscript.

We thank the reviewer for the positive comments and the recommendation to publish our manuscript

Reviewers' Comments:

Reviewer #2:

Remarks to the Author:

Thank you for the efforts to ensure the manuscript reports the interaction of the proteins of interest and not of the perhaps unknown effects of the GST tag

The problem is increased by many omitted information

For example, while one can dig out reference #26 to hopefully find out the details about the tag and any additional linker residues, it would be helpful to have these stated on line 393 in the methods section

Likewise for the GST tagged beta-catenin: please describe the construct in line 404 of the methods section and please provide information about the linker and any additional non-beta-catenin residues

Importantly, please refer to experimental beta-catenin as GST-beta-catenin every time and not just as beta-catenin so that every Figure can be interpreted as being the fusion

Perhaps even same for alpha-catenin, please state His(x)-alpha-catenin instead of "alpha-catenin"

The Rimm et al. (1995) PNAS paper shows that GST-beta-catenin does sediment in the absence of F-actin and the PNAS paper point was that there is no apparent increase sedimentation of GST-beta-catenin in the presence of F-actin (lanes 2 versus 4 of their Figure 1)

One should keep in mind that these experiments were done 26 years ago, and they are not showing any quantification of the pellets

Supplementary Figure 14d is unclear due to insufficient information: please state that (and how much) GST-beta-catenin was bound to the nitrocellulose smeared cover slides and the control (GST alone) is maybe not needed since they get no binding of GST-beta-catenin

With regards to their 34 nM affinity, the constructs are not really comparable:

1. The Koslov et al. (1997) paper determined the interaction to have a KD of 100 nM for untagged beta-catenin

2. The 2014 Pokutta et al. paper found a 20 nM affinity for untagged alpha NI-catenin (= residues 1-264) or alpha NII-catenin (= 48 aa insert between 810-811) for binding to untagged beta-catenin

But perhaps it is ok to believe that these values are "in good agreement"

The added supplementary discussion entitled "possible effect of the GST tag in beta-catenin" needs to be removed in the absence of showing the data

Since there is a clear biological function of monomeric alpha-catenin that is distinct from dimeric alpha-catenin and with the GST-tagged beta-catenin, alpha-catenin bound to GST-beta-catenin is artificially in proximity because of the dimeric beta-catenin that does not exist in cells as a dimer

In contrast, in cells, the monomeric alpha-catenin form is in equilibrium with the dimeric alpha-catenin form

Please have a look at 1dow (alpha-catenin, yellow; beta-catenin magenta; alpha-catenin C-alpha residue 57, red; beta-catenin C-alpha residue 118, blue) superimposed onto 2z6g (white) where the

N-termini (blue) indicate where the additional residues and dimeric GST would be (and the entire C-terminal alpha-catenin domain is not shown)

Without stating that some lanes are not part of the experiment and just show the purity of the proteins (right?) one would wrongly conclude that:

1. Supplementary Figure 11a shows a significant amount alpha-catenin in the absence of actin in the pellet
2. Supplementary Figure 11b and 11e shows a significant amount of actin in the supernatant

The beta-catenin control (centrifuged in the absence of actin) is still missing

We thank again the reviewer for his/her comments. We have carefully addressed all points and we include below a point-by-point response to the reviewer's comments. Moreover, we highlighted all changes in yellow in the manuscript text file and in the supplementary information.

REVIEWER COMMENTS

Reviewer #2 (Remarks to the Author):

Thank you for the efforts to ensure the manuscript reports the interaction of the proteins of interest and not of the perhaps unknown effects of the GST tag

We thank the reviewer for the appreciation of our work.

The problem is increased by many omitted information

For example, while one can dig out reference #26 to hopefully find out the details about the tag and any additional linker residues, it would be helpful to have these stated on line 393 in the methods section

We have better specified the vector name and brand and added the required information in the methods section (line 400-404), "The plasmid encodes the protein fused with a 6x His-tag at the N-terminus with a linker sequence containing a thrombin cleavage site and restriction sites for NdeI and NheI (amino acid sequence of the His tag and linker: "MGSSHHHHHSSGLVPRGHMAS")."

Likewise for the GST tagged beta-catenin: please describe the construct in line 404 of the methods section and please provide information about the linker and any additional non-beta-catenin residues

We have added the required information (line 414-416), "β-catenin is fused with a GST tag at the N-terminus with a linker sequence containing a thrombin digestion site (amino acid sequence of the linker: "SDLVPRGS")"

Importantly, please refer to experimental beta-catenin as GST-beta-catenin every time and not just as beta-catenin so that every Figure can be interpreted as being the fusion

Perhaps even same for alpha-catenin, please state His(x)-alpha-catenin instead of "alpha-catenin"

We understand the referee's request to be clear about the presence of the GST and His tags in the proteins. In the article most of the times we refer to the alpha-beta-catenin heterodimer, and it would be difficult to read as an His(6x)-alpha-catenin-GST-beta-catenin heterodimer. So, to address the referee's concern, we made even more clear in each figure caption of the main article that the two proteins are present as fusion with the respective tags. In particular:

- in the caption of Figure 1 (line 136-137), we added this sentence: "A single α -catenin homodimer was composed by two His(6x) tagged α -catenins (see methods)".
- in the caption of Figure 2 (line 181-183), we added this sentence: "The single heterodimer is composed by a single monomeric His(6x) tagged α -catenin attached onto a layer of GST-tagged β -catenin (see methods)."
- In the caption of Fig. 3, we added this sentence (line 239-241): "Multiple heterodimers are composed by monomeric His(6x) tagged α -catenins attached onto a layer of GST-tagged β -catenin (see methods)."

Moreover, also in the Supplementary Information we have specified that the two proteins are present as fusion with the respective tags:

- Section " α -catenin-actin pull-down assay": line 157 and caption of Supplementary Fig. 11.
- Section " α -catenin- β -catenin pull-down assay": line 177-178 and caption of Supplementary Fig. 12.
- Section "flow cell assay": line 220 and caption of Supplementary Fig. 14 (line 255).
- Section "non-specific interactions": line 329

The Rimm et al. (1995) PNAS paper shows that GST-beta-catenin does sediment in the absence of F-actin and the PNAS paper point was that there is no apparent increase sedimentation of GST-beta-catenin in the presence of F-actin (lanes 2 versus 4 of their Figure 1)

One should keep in mind that these experiments were done 26 years ago, and they are not showing any quantification of the pellets

We agree with the referee that in the paper by Rimm et al. there is no quantification of the pellets. Their paper qualitatively confirms our single molecule and flow cell control experiments in which we do not observe non-specific interactions between GST-beta-catenin and actin. Moreover, we cited the Rimm et al. paper because it stated that "the results of each assay were then confirmed by using peptides in which the GST sequences had been removed by proteolysis with thrombin" (Fig.1 caption) and "These findings were not an artifact of the GST fusion sequence on the recombinant peptides, since proteolytic removal of GST had no effect on any of these activities, nor did the other GST-containing peptides induce the low-speed sedimentation of actin." (page 8815). We modified our reference to the Rimm et al. paper in the Supplementary Discussion section to better describe the main point of the

paper, as suggested by the referee (line 361-363): “Although previous studies indicate that the GST tag does not introduce non-specific interactions with actin², we directly tested whether the GST tag might introduce non-specific interactions with actin, α -catenin, or both.”

Supplementary Figure 14d is unclear due to insufficient information: please state that (and how much) GST-beta-catenin was bound to the nitrocellulose smeared cover slides and the control (GST alone) is maybe not needed since they get no binding of GST-beta-catenin

We modified the caption of Supplementary Fig. 14d indicating that GST-beta-catenin was bound to the nitrocellulose smeared coverslide and we specified its concentration (line 255). We agree with the referee that a control experiment with GST alone is not needed.

With regards to their 34 nM affinity, the constructs are not really comparable:

1. The Koslov et al. (1997) paper determined the interaction to have a KD of 100 nM for untagged beta-catenin
2. The 2014 Pokutta et al. paper found a 20 nM affinity for untagged alpha N-catenin (= residues 1-264) or alpha NII-catenin (= 48 aa insert between 810-811) for binding to untagged beta-catenin

But perhaps it is ok to believe that these values are “in good agreement”

The added supplementary discussion entitled “possible effect of the GST tag in beta-catenin” needs to be removed in the absence of showing the data

The Koslov et al. (1997) paper determined kD of 100nM between untagged full-length alpha-E-catenin and untagged full-length beta-catenin (however no error is reported on this value). It is true that Pokutta et al. (2014) determined kD for alpha-N-catenin, as reported by the referee, but they also measured $kD = 23.4 \pm 3.7$ between untagged full-length alpha-E-catenin and untagged full-length beta-catenin (see Table 3 of their paper). Therefore, the constructs of Koslov et al. and of Pokutta et al. are comparable. We agree with the referee that the kD values are quite different, but with no error on the 100 nM value it is difficult to say whether the two values are compatible or not. Our experiments show that alpha-catenin binds to GST-beta-catenin with $kD = 35 \pm 13$ nM, which is compatible (within the error) with the 23.4 ± 3.7 nM value reported by Pokutta et al. and, again, we cannot say whether it is compatible with the 100 nM value reported by Koslov et al. Therefore, although we cannot exclude a possible influence of the GST tag on this interaction, we do not see a significant effect compared to previous studies. Following the referee’s comment, we modified the supplementary discussion entitled “possible effect of the GST tag in beta-catenin” showing the data from Koslov et al and Pokutta et al. Moreover, we

changed the expression “in good agreement” with a more extended sentence to better explain the comparison with previous results and limiting the comparison to the data of Pokutta et al., who quantified the uncertainty on their measurement (lines 191-194 and 377-382 of Supplementary Information).

Since there is a clear biological function of monomeric alpha-catenin that is distinct from dimeric alpha-catenin and with the GST-tagged beta-catenin, alpha-catenin bound to GST-beta-catenin is artificially in proximity because of the dimeric beta-catenin that does not exist in cells as a dimer

In contrast, in cells, the monomeric alpha-catenin form is in equilibrium with the dimeric alpha-catenin form

Please have a look at 1dow (alpha-catenin, yellow; beta-catenin magenta; alpha-catenin C-alpha residue 57, red; beta-catenin C-alpha residue 118, blue) superimposed onto 2z6g (white) where the N-termini (blue) indicate where the additional residues and dimeric GST would be (and the entire C-terminal alpha-catenin domain is not shown)

We thank the referee for this comment. We experimentally checked, also thanks to previous reviewer’s comments, the interactions between GST-beta-catenin and alpha-catenin and we did not observe significant deviations from previous data on unlabeled beta-catenin. We cannot exclude that the GST tag might have an effect on the interaction with alpha-catenin, although this is not detected by our experiments. However, we believe that discussing the structural details of the positioning of the GST tag at the N-terminus of beta-catenin and its possible influence on the interaction with alpha-catenin is out of the scope of the present work.

Without stating that some lanes are not part of the experiment and just show the purity of the proteins (right?) one would wrongly conclude that:

1. Supplementary Figure 11a shows a significant amount alpha-catenin in the absence of actin in the pellet
2. Supplementary Figure 11b and 11e shows a significant amount of actin in the supernatant

In the revised paper we have specified the content of each lane in the figure caption to avoid misunderstandings (caption of Supplementary Fig. 11).

The beta-catenin control (centrifuged in the absence of actin) is still missing

In the first set of pull-down experiments, we tested the interaction between alpha-catenin and actin, in the absence of beta-catenin. In the second set of pull-down experiments, we tested the interaction between beta-catenin (attached to micron sized beads) and alpha-catenin, in the absence of actin.

Therefore, we do not see the point in making a control of beta-catenin centrifuged in the absence of actin. Maybe the referee intended beta-catenin beads centrifuged in the absence of alpha-catenin, however this control experiment is not necessary because it is clearly visible from Supplementary Fig 12c, lane 4, that at the lowest alpha-catenin concentration used (2 nM), no band corresponding to GST-beta-catenin (about 115 KDa MW) is detectable in the pellet.